# Needs Hierarchy for Public Service Facilities and Guidance-Control Programming in Small Chinese Towns Influenced by Complex Urbanization of Residents: The Evidence from Zhejiang

**Zhi Qiu** [1,2,†] 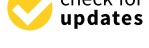**, Yue Wang** [1,3,†] **, Jie Wang** [1,2] **, Zhu Wang** [1] **and Yi Zhou** [4,*]

1   Institute of Architectural Design and Theoretical Research, Zhejiang University, Hangzhou 310058, China; qiuzhi0710@zju.edu.cn (Z.Q.); wangyue0801@zju.edu.cn (Y.W.); wangjie@zju.edu.cn (J.W.); 13325716910@163.com (Z.W.)
2   Center for Balance Architecture, Zhejiang University, Hangzhou 310027, China
3   The Architectural Design and Research Institute of Zhejiang University Co., Ltd., Hangzhou 310027, China
4   Zhejiang Design Institute of Water Conservancy & Hydro-Electric Power Co., Ltd., Hangzhou 310002, China
*   Correspondence: leept@sina.com; Tel.: +86-138-1910-9877
†   These authors contributed equally to this work and should be considered co-first authors.

**Abstract:** Due to the rapid urbanization of China, residential areas and residents in small towns exhibit dual and complex urbanization patterns and characteristics. The "one-size-fits-all" approach to programming public service facilities has led to inefficient use of idle facilities and the frequent use of facilities that are not integrated into the public service system. Therefore, an investigation of programming logic based on residents' needs within the complex urbanization patterns and characteristics of small towns is urgently required. This study distinguishes between rural and nonrural facilities, using the Kano model to evaluate residents' hypothetical satisfaction with the provision of facilities or lack thereof. Based on the "rural and nonrural" facility need coupling model, four stages of urbanization were identified. Using the Better–Worse coefficient and the chi-square test, this study analyzes residents' cognitive needs and population attributes across various stages of urbanization. Moreover, the study examines the "residential area–resident facility" matching relationship. Ultimately, a "required + optional item" public service facility guidance-control system based on the matching of human–land urbanization is proposed to improve the current programming system for public service facilities. This system provides a theoretical basis for improving the public service level in small towns and ensuring optimally relevant regulations.

**Keywords:** small towns; public service facilities; programming; suitability; human–land urbanization patterns and characteristics

## 1. Introduction

The development patterns of cities and rural spaces are becoming ubiquitously similar across the globe. In the three-level "city–town–village" structure, small towns are widely regarded as vital nodes connecting cities and villages [1]. They not only offer essential employment, basic services, and public transportation to residents and nearby rural settlements but also serve as the economic hinterlands of cities, attracting relocated businesses and developing supporting industries [2]. Due to the advanced processes of industrialization and urbanization, small towns in developed countries are usually urbanized areas that do not include farmland or typical rural production activities [3]. They tend to be residential units at a smaller scale than cities and are defined by standardized indicators, such as population size and density [4], which are similar to those used to define cities. China continues to undergo urbanization. Some small towns have already shown standardized and homogeneous features similar to those of cities, while others serve

as spatial support for nearby urbanization [5,6]. The impetus of rural industrialization and the influx of enterprises have led to the expansion of township construction land [7], which corresponds to a reduction in rural residential areas and available space [8]. The evolution of urban–rural relationships has thus resulted in the emergence of "semi-urban, semi-rural" characteristics in small towns. The geographer McGee referred to the pattern formed by the flow and reconfiguration of these factors between urban and rural areas as "Desakota" (derived from the Indonesian words "desa", meaning village, and "kota", meaning town) [9], while other scholars have referred to this pattern as the urban–rural continuum [10]. In a township's construction land, there are various types of residential areas. These include urban communities and areas with urban features, such as apartment resettlement areas [11]. In addition, there are residential areas with rural features, such as rural settlements [12] and homestead resettlement areas [13]. As urbanization occurs, farmers undergo a transition from a rural lifestyle to an urbanized one, involving the stages of "local nonrural, passive urbanization, and gradual urbanization" [14]. Consequently, residents of small towns enter different stages of urbanization; there is a mismatch in the degree of urbanization between residential areas and residents. For instance, some residents living in rural settlements near towns will experience a high degree of urbanization due to long-term environmental infiltration. As a result, the speed of urbanization among residents outpaces improvements in their living spaces [15]. However, some residents exhibit characteristics that are inconsistent with the environment due to their low adaptability after environmental transformation [16]. For example, some residents have converted their living spaces into rural-style living quarters and have even repurposed them for production activities, which is incongruous with the intended use of these areas [17].

This complex human–land urbanization pattern has intensified the challenges associated with constructing a living environment in small towns, which is reflected in the programming of public service facilities. As a space carrier supporting the production and living needs of residents, the programming of public service facilities is crucial in improving the quality of the corresponding living environment. Although some small towns in China have shown urban-like characteristics, a considerable number of small towns have "semi-urban, semi-rural" characteristics. Currently, the programming of public service facilities in small towns is based on urban-like standards, such as the national "Standard for Planning of Towns" (GB50188-2007) and relevant local regulations. This approach to facility programming, which focuses on achieving the "full coverage of indicators", fails to consider the classification and guidance of residential areas based on urbanization patterns, resulting in the inefficient usage of idle and frequently used facilities that are not included in the public service system [18]. This has a negative impact on the livability of the environment and the surrounding rural areas [19], leading to population outflow and the issue of rural hollowing out.

In the process of constructing a living environment in villages and towns, several countries advocate a "place-based approach" [20] and a development model that considers grassroots forces [21] rather than blindly pursuing universal standards. For example, the Commission of European Communities suggested in its "Green Paper on Territorial Cohesion" that strategies for development should be based on the inherent characteristics of the territory and leverage the unique assets of the area, such as its physical, human, and social capital as well as its natural resources [22]. Considering the diversity of rural areas, the European Commission proposed the LEADER series of programs (Links Between Actions of Rural Development) to address local needs through region-specific development strategies [23]. China is a vast country with a variety of small towns; the traditional top-down political and governance systems [24] cannot fully implement the standardized "one place, one policy" approach in facility programming. Therefore, introducing the concept of "control and guidance" is necessary. To avoid the misallocation of resources and since residents are the direct users of these facilities, exploring the principle of programming based on residents' needs is crucial. This is applicable to different residential area patterns such that the programming of facilities matches the dual human–land urbanization pattern

under top-down control. By implementing optimal facility programming, this study aims to enhance the quality of human settlements and promote the sustainable development of small towns, thus achieving parity in public services between urban and rural areas.

## 2. Methods

### 2.1. Cognitive Paths of Resident Types in Diverse Urbanization Patterns

Due to the neglect of urbanization patterns of residential areas and the practice of solely exploring residents' needs in different urbanization stages from the perspective of human urbanization patterns, decision makers must identify the complex urbanization characteristics of different population groups and select facility types based on the diverse needs of residents within the practice of programming in small towns. The operational feasibility of the outcomes and guidance of relevant decision-making departments may be relatively low. Therefore, this study first focuses on residential areas, exploring the types and characteristics of residents' urbanization according to the urbanization patterns of residential areas, in order to develop a facility-programming system tailored to different residential areas.

2.1.1. Cognitive Perception of the Types of Urbanization Patterns in Residential Areas

From the perspective of spatial layout, urban community areas, apartment resettlement areas, and homestead resettlement areas all present planned features, while rural settlements demonstrate a self-organized form. In terms of residential form, there is not a significant difference between urban community areas and apartment resettlement areas; both comprise multistoried or high-rise structures. In contrast, homestead resettlement areas retain low-rise courtyard forms similar to those found in rural settlements (Table 1). Based on spatial analysis, research has confirmed that rural settlements and homestead and apartment resettlement areas are gradually becoming more urbanized. Homestead and apartment resettlement areas exhibit "rural-like" and "urban-like" urbanization characteristics, respectively [25]. Based on the trends of urbanization features present in the four types of residential areas, this study categorizes residential areas into two types: "urban-biased residential areas" (i.e., urban and urban-like areas), such as urban community and apartment resettlement areas, and "rural-biased residential areas" (i.e., rural and rural-like areas), such as rural settlements and homestead resettlement areas.

**Table 1.** Illustrative diagrams of spatial layouts and residential forms of different residential areas.

| Urban-Biased Residential Areas | | Rural-Biased Residential Areas | |
|---|---|---|---|
| Urban community area | Apartment resettlement area | Homestead resettlement area | Rural settlement |

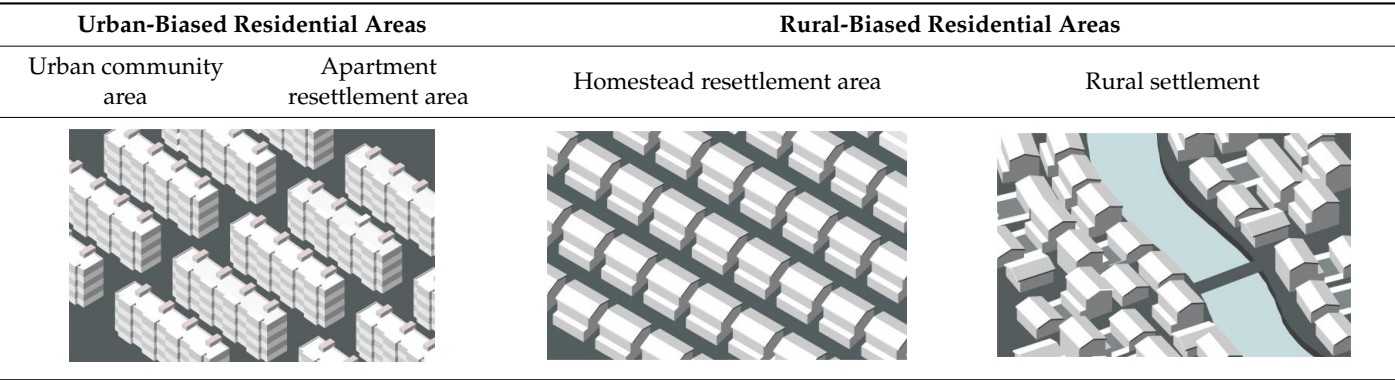

2.1.2. Analysis of Urbanization Characteristics Based on Residents' Facility Needs

There are multiple academic approaches to distinguishing an individual's urbanization stage. Some studies have observed a correlation between population attributes and individuals' urbanization stages, indicating that attributes such as household registration status, gender, years of education, and age have varying degrees of impact on an individual's level of urbanization [26]. However, an individual's urbanization stage is not only determined by their surface-level population attributes but also by their "inner" identifica-

tion and adaptation to urban life and production [27]. Therefore, this study uses the degree of residents' needs for different types of facilities as the main criterion for distinguishing individuals' urbanization stages.

During nearby urbanization, residents experience transformations in their living environments and modes of living and producing, such as relocating from rural areas to multistoried buildings and resettling in a homestead [28]. In this context, residents often choose to contract their land, shifting away from agriculture as a primary source of income. Some residents, due to their strong adaptability to their environments, adjust to the production modes and lifestyles available in urban areas; they have a higher need for certain living service facilities [29]. In addition, some residents are still undergoing adaptation to their identity-related and environmental changes. They continue to hold traditional ideological beliefs and rely on living service facilities, such as ancestral halls and cultural auditoriums to hold gatherings [30]. Furthermore, they rely on the community for economic income [31] and depend on productive services, such as materials and machine shops, for their agricultural economic activities. Thus, the varying degrees of the requirement for different types of facilities among residents reflect the differences in their stages of urbanization. By summarizing the characteristics of these phenomena, this study classifies facility types into rural and nonrural facilities. Rural facilities are unique to rural areas and continue to be relied upon by less-urbanized residents, while nonrural facilities are available in urban and rural areas. Subsequently, this study analyzes residents' different urbanization stages based on their varying levels of requirement for the two types of facilities, classifies the residents accordingly, and establishes a "residential area–resident facility" matching relationship.

Construction of "Rural and Nonrural Facility" Library: To clarify the scope of this research, in this study, a public service facility and a library were constructed for rural and urban communities. The library was created by reviewing national and local regulations, policy guidelines, and relevant literature and by conducting field observations of frequently used facilities that had not yet been incorporated into the public service facility system. Then, we compared the library with the "Standard for Urban Residential Area Planning and Design" (GB50180-2018) and selected facility types that are only available as rural facilities, while those available in both urban and rural areas were selected as nonrural facilities (Table 2).

**Table 2.** "Rural and nonrural facility" library.

| Types | | Facilities |
|---|---|---|
| Nonrural facilities | Education | Kindergarten, primary school, junior school, senior high school, training institution, adult vocational and technical school |
| | Medical | Clinic, hospital, specialized clinic, emergency medical site (nucleic-acid-testing site), Hugh sanatorium |
| | Cultural and Sports | Outdoor sports ground, indoor sports venue, cultural activity center, exhibition hall |
| | Social Welfare | Home aged care service center, nursing home, service station for the disabled, social welfare institute, villager canteen |
| | Administration | Neighborhood (village) service center |
| | Commerce | Grocery, agricultural market, food store, sales and maintenance department of non-motor vehicles, hardware store, pharmacy, department store, book and video store, cultural goods store, express/postal service station, self-service cabinet, bank, telecom business hall, insurance institution, restaurant/teahouse, beauty and hair salon, photographic studio, public bathroom |
| | Infrastructure | Bus station, motor vehicle parking lot, public toilet, garbage collection point |
| | Productive Service (Tourism, etc.) | Tourist reception center, hotel/homestay, tourist souvenir shop |

**Table 2.** *Cont.*

| Types | | Facilities |
|---|---|---|
| Rural facilities | Productive Service (Agriculture, etc.) | Agricultural machine shop, agricultural materials shop, grain-drying site, agricultural product storage station, product sales site, agricultural product acquisition station, farm tool storage station, agricultural technical education station, agricultural cooperative |
| | Cultural and Sports | Cultural auditorium, ancestral hall, temple, film-screening venue, broadcasting station |

Construction of the Model Framework for Residents' Urbanization Characteristics: After classifying the residents based on their levels of recognition of the types of facility, a "rural and nonrural" facility need coupling model framework was constructed to reflect the differentiation of residents' urbanization stages (Figure 1).

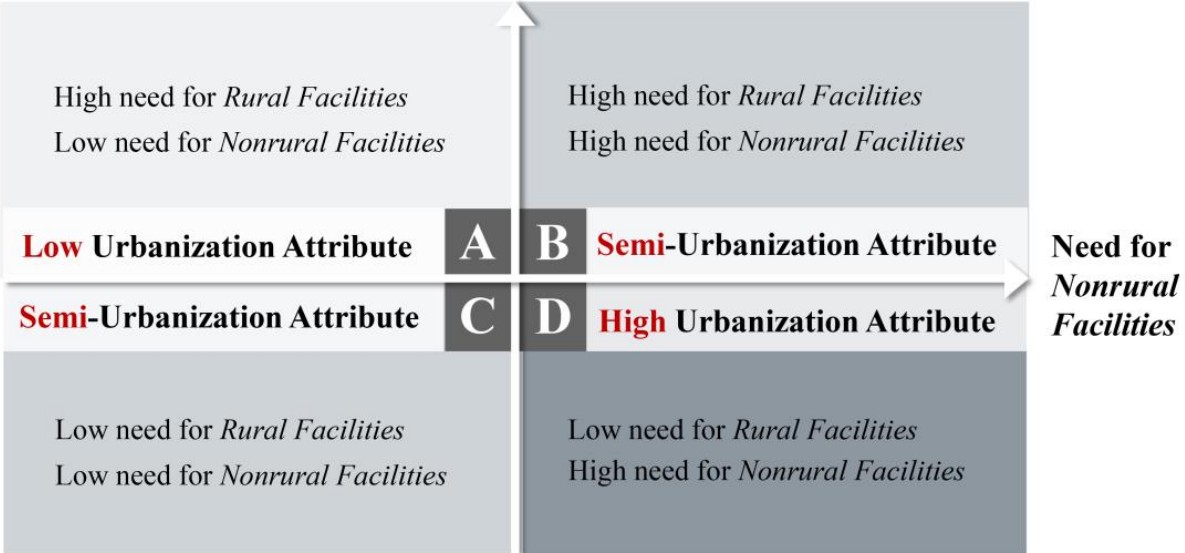

**Figure 1.** Framework of the "rural and nonrural" facility needs coupling model.

Related studies have shown that the cognitive characteristics of residents' production and living needs in small-town apartment resettlement areas reflect their levels of urbanization [32]. Residentswith a high standard of living and with low production requirements belong to the urban group. Hence, the degree of need for community production is a factor that reflects residents' level of urbanization. However, the indication system for their living needs includes rural and nonrural facilities, as defined in this study. Therefore, this system fails to account for the important fact that rural living service facilities tend to be used by residents with low levels of urbanization. Thus, this study further refines the model framework for residents' urbanization characteristics, categorizing rural facilities into dimensions of production and living needs, and suggesting that residents who present high levels of requirement for rural facilities in terms of either production or living needs should be classified as those with a "high need for rural facilities."

Matching Analysis of the Residents' Urbanization Characteristics: Using the chi-squared test, we conducted a significance test on the percentages of individual characteristics among different urbanization stage groups. The significant differences between the four categories of urbanization characteristics and population attributes were analyzed. The urbanization characteristics of residents living in various urbanization patterns of residential areas were described, providing support for menu-oriented facility programming.

### 2.2. The Facility Programming Path Based on Residents' Needs

#### 2.2.1. Establishment of a Guidance and Control Hierarchy Based on Needs Theory

"Need" is the internal force that drives individuals to seek and utilize specific objects [33]. It represents the pre-motivational factor driving human behavior [34] and is the key driver of facility usage. Considering the restrictions imposed by policies and funding in villages and towns [35], the establishment of facilities cannot be achieved overnight. Therefore, prioritization based on the urgency of a need is necessary [36–38]. In terms of the hierarchy of needs, American psychologist Abraham Maslow proposed a theory encompassing physiological, safety, social, esteem, and self-actualization needs [39]. Maslow believed that lower-level needs must be adequately met before higher-level needs can be satisfied [40] and that this process is irreversible [41]. Clayton P. Alderfer, an American psychologist, developed the Existence–Relatedness–Growth (ERG) Theory, building upon Maslow's Hierarchy of Needs [42]. Both theories suggest that once lower-level needs are met, there is a desire for higher-level needs. However, the ERG Theory expands upon this idea by proposing that different levels of needs can coexist [43,44] and multiple needs can simultaneously act as motivators. Moreover, the ERG Theory incorporates the idea of "frustration-regression", which suggests that when higher-level needs are not met, individuals may regress to lower-level needs [45,46]. In the allocation of public service facilities, not all high-level needs can be met. Therefore, low-level needs should be controlled through "rigid" measures, while high-level needs should be guided through "flexible" measures. A "required + optional item" programming model should be adopted to respond to the structural characteristics of residents' public service facility needs (Figure 2). Hence, facility implementation should prioritize satisfying the mandatory requirements (low-level needs) before the optional requirements (high-level needs). To identify the overall need hierarchy of residents with different urbanization levels in each analyzed residential area, an individual need structure is identified. Then, based on certain principles, this study integrates the needs of the residents and, subsequently, obtains a programming guidance-control system for different residential areas.

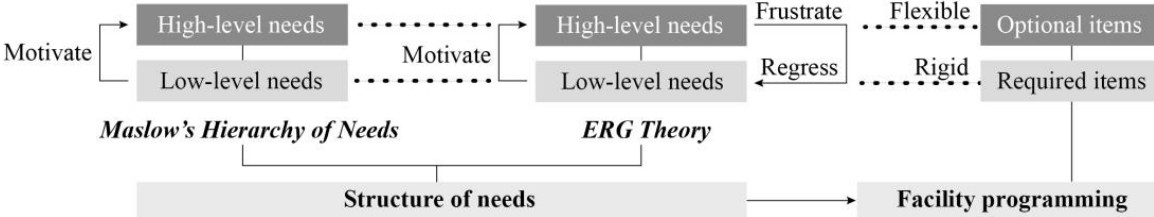

**Figure 2.** "Required + Optional Item" Programming Model Based on Needs Theory.

#### 2.2.2. Identification of Needs Structure Characteristics Based on the Kano Model

As this study explores the issue of whether a facility should be allocated in different "human–land urbanization patterns", it focuses on examining people's attitudes toward the provision of a given facility. Therefore, satisfaction was selected as the measurement indicator. The measurement methods for satisfaction can be divided into two categories: experiential and hypothetical evaluation. The former measures overall satisfaction using standardized scales, such as SERVQUAL [47], or through methods with their own indicator systems, such as the analytical hierarchy process (AHP) [48]. Additionally, there are methods that assess individual attribute satisfaction, such as importance performance analysis, which segments attributes into four quadrants [49]. To develop the programming guidance-control system, facilities that are not yet available but may constitute part of residents' potential needs should be considered in addition to those included in the public service system. Accordingly, this study aims to employ a "hypothetical" method to measure satisfaction in order to identify the characteristics of the needs structure. The Kano model is precisely such a method, which was proposed by Professor Kano Noriaki of the Tokyo Institute of Technology in 1984 [50]. This model has been widely applied in the fields of

service development and improvement [51,52]. It suggests that not every service provision or lack thereof will increase or decrease, respectively, users' satisfaction with the service in question [53].

There are typically two types of satisfaction assessment questions asked of users: positive and negative. Positive refers to the satisfaction of the respondent when the service attribute is present or functioning properly (Functional). Negative refers to the satisfaction of the respondent when the service attribute is lacking or experiencing malfunction (Dysfunctional) [54]. The responses to positive and negative questions are correlated, reflecting five attributes: must-be (M), one-dimensional (O), attractive (A), indifferent (I), and reverse quality (R) (Figure 3). The Kano evaluation table (Table 3) maps the responses to positive and negative questions to cognitive attributes. For example, if a respondent chooses "Dissatisfied" for dysfunctional and "It should be that way", "I am indifferent", or "I can live with it" for functional, the corresponding cognitive attribute in the Kano evaluation table is the Must-be quality attribute (M). Compared to the traditional linear Likert scale method, the Kano model can identify residents' satisfaction with the provision or non-provision of a certain facility, thus avoiding respondents' cognitive bias toward providing positive responses [55]. In particular, the Kano model effectively conveys the desired "urgency" of facility construction, providing valuable support for the classification of facilities based on residents' actual needs, especially in situations where resources are limited.

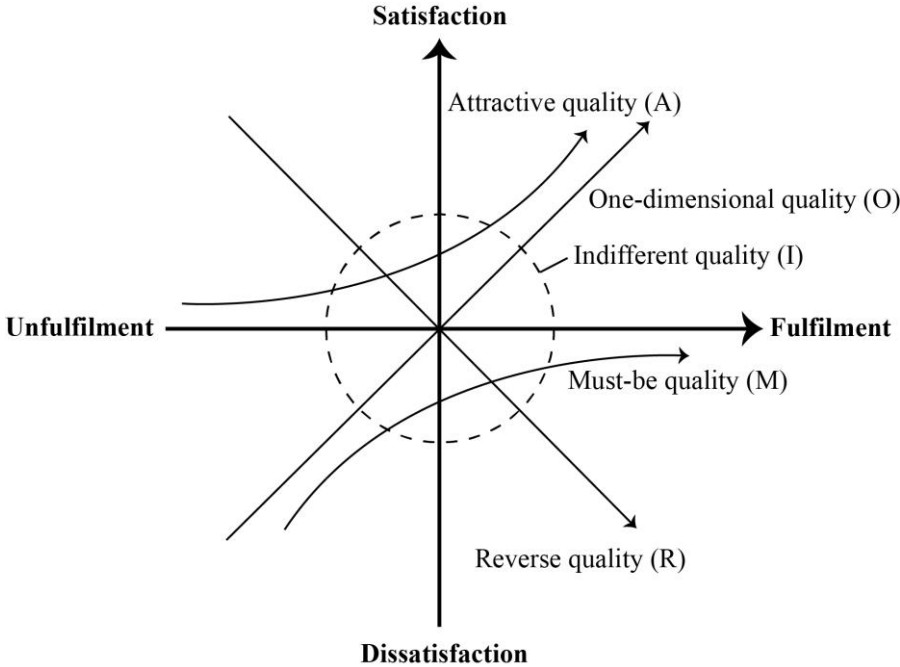

**Figure 3.** Kano's model of quality attributes [56].

**Table 3.** Kano evaluation table.

| Criteria/Attributes | | Dysfunctional | | | | |
|---|---|---|---|---|---|---|
| | | Satisfied | It Should Be That Way | I Am Indifferent | I Can Live with It | Dissatisfied |
| **Functional** | Satisfied | Q | A | A | A | O |
| | It should be that way | R | I | I | I | M |
| | I am indifferent | R | I | I | I | M |
| | I can live with it | R | I | I | I | M |
| | Dissatisfied | R | R | R | R | Q |

Each level is defined as follows:

- Must-be quality (M): Facilities that are essential to the lives of residents. Providing these facilities in the community may not increase residents' satisfaction; however, without such facilities, residents will be extremely inconvenienced.
- One-dimensional quality (O): Facilities that residents expect to have. The provision of these facilities will satisfy the residents, while the absence of such facilities will make the residents feel dissatisfied; however, they can tolerate such an absence.
- Attractive quality (A): Facilities that are unexpected and will pleasantly surprise residents. Such facilities are beyond residents' expectations and do not affect their normal lives if they are unavailable; however, if provided, they can greatly improve satisfaction and quality of life.
- Indifferent quality (I): Facilities that are not a concern for residents. Whether such facilities are provided or not has a minimal effect on the residents' quality of life.
- Reverse quality (R): Facilities that residents do not want.

The research flow chart is shown in Figure 4.

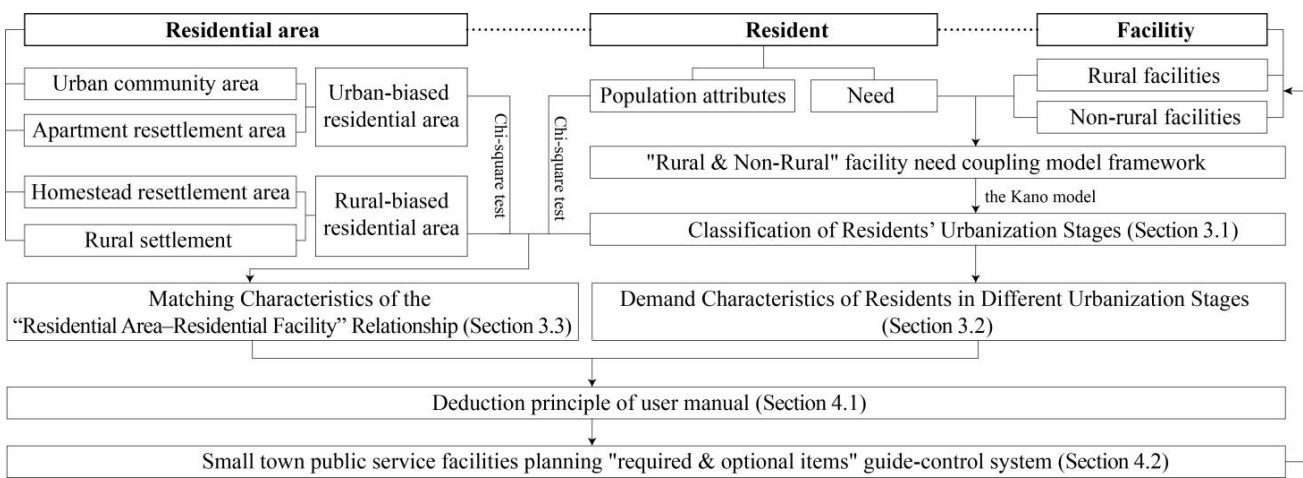

**Figure 4.** Research flow chart.

*2.3. Investigation Method*

2.3.1. Sample Selection

This study aimed to select a sample of small towns that exhibit the dual complexity of human–land urbanization. Due to the development trend of the industrial economy over the past decade, S town in Jiaxing City, Zhejiang Province, has been continuously developing industrial parks in its northern and eastern regions, resulting in a gradual expansion of township construction land into urban areas (Figure 5). Currently, the township construction land includes various types of residential areas, such as rural settlements, homestead resettlement areas, urban community areas, and apartment resettlement areas (Figure 6). Differences in urbanization characteristics are evident among residents in the different residential areas. The majority of the residents living in rural-biased residential areas have not yet converted their household registration and continue to rely on agriculture as their primary source of income. Meanwhile, most residents living in urban-biased residential areas have fully transitioned away from agriculture, and displaced farmers have found employment in local industrial parks. Overall, the residential areas and residents within S Town exhibit complex urbanization characteristics, making the town an appropriate sample for the study.

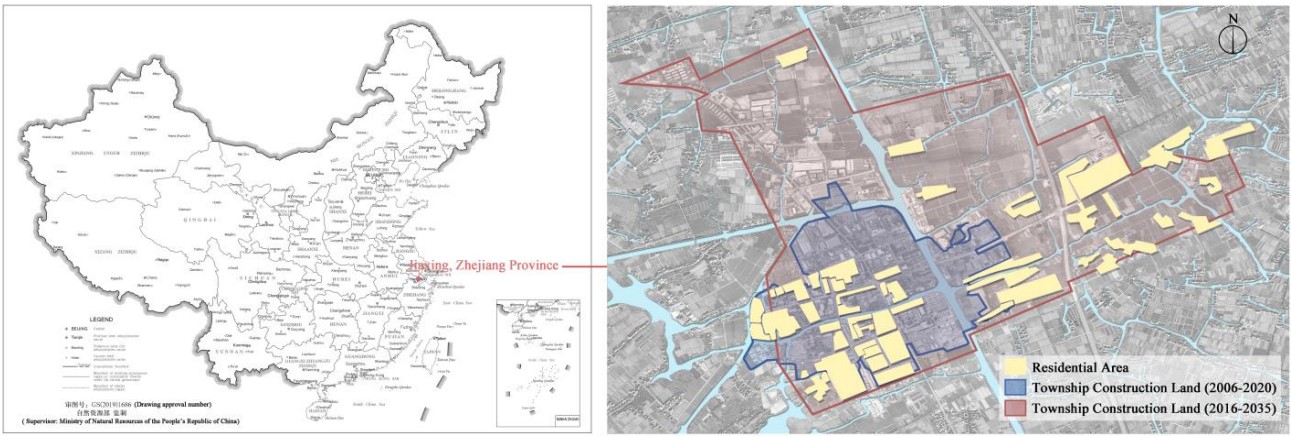

**Figure 5.** Changes in the scope of township construction land.

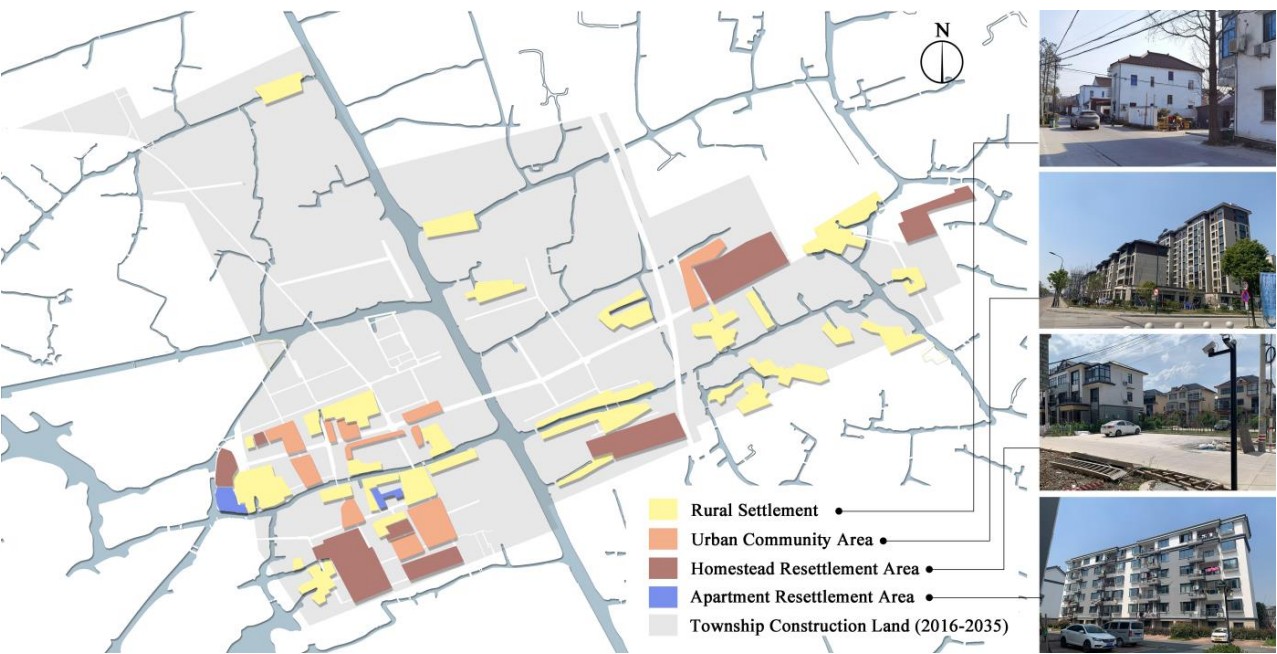

**Figure 6.** Plane distribution of various residential area types.

2.3.2. Questionnaire Design and Investigation Process

The survey questionnaire comprises two sections: gathering basic demographic information about residents and conducting a satisfaction survey using the Kano model. The basic demographic information serves as a tool for evaluating population attributes across various stages of urbanization. It encompasses details such as residential area, gender, age, registered permanent address, educational level, religious beliefs, household responsibilities, employment status, annual household income, marital status, and living arrangements with family members, among others. The satisfaction survey specifically targets each facility and incorporates a combination of positive and negative questions. An illustrative example of the questionnaire is provided in Table 4.

Before conducting the investigation, it is important to provide training to the investigators and simulate investigate scenarios, thereby mitigating potential information collection errors resulting from subjective interpretation biases. As this study aims to identify the genuine needs of residents within the context of human–land urbanization patterns, it was also essential to clarify during the training that the interviewees should have been registered residents who had been residing in the town in question for a long period. Temporary

residents or those who commute to the town for work do not fall within the scope of this investigation. The investigation adopts a semi-structured interview approach to avoid potential biases in questionnaire comprehension due to factors such as educational level. With the coordination and assistance of government officials, the investigators visited residents' homes, explained the questionnaire to them, and collected data on an individual basis. This approach helped to ensure the validity of the questionnaire data to a certain extent.

**Table 4.** Example of satisfaction questionnaire based on the Kano model.

| Type | Facility and Service | If There Is Such a Facility Nearby | | | | | If There Is No Such Facility Nearby | | | | |
|---|---|---|---|---|---|---|---|---|---|---|---|
| | | Satisfied | It Should Be That Way | I Am In-different | I Can Live with It | Dissatisfied | Satisfied | It Should Be That Way | I Am In-different | I Can Live with It | Dissatisfied |
| Medical | Clinic | | | | | | | | | | |
| | Hospital | | | | | | | | | | |
| … … | … … | | | | | | | | | | |

To ensure adequate representation of the overall population, both regular days and special days (weekdays: 7–8 July 2022; weekends: 9–10 July 2022) were selected, and investigators were assigned to conduct surveys in each residential area. The study balanced the data collection quantity between rural- and urban-biased residential areas. A total of 260 questionnaires were distributed. After excluding questionnaires with missing information or incomplete responses, 252 valid questionnaires were collected, comprising 112 and 140 responses from urban- and rural-biased residential areas, respectively.

## 3. Results

### 3.1. Classification of Residents' Urbanization Stages

This study used the Kano evaluation table to determine the cognitive attributes with respect to facilities among individual residents. To measure the participants' levels of recognition of rural and nonrural facilities, attribute values were assigned (must-be = 1, one-dimensional = 2, attractive = 3, indifferent = 4, and reverse = 5). The average value for the different types of facility attributes was calculated for each individual, and a violin plot was created (Figure 7) to make a preliminary judgment on the data distribution. There were distinct peaks and valleys in the three categories, indicating that the residents' perception of these facilities had a certain degree of clustering, further confirming the hypothesis of population classification and demonstrating the feasibility of clustering. As a violin plot can visually determine the number of clustering groups, the K-means clustering method was used to cluster the cognitive attributes of the three types of facilities into high and low levels (Figure 8). As a result, four groups were identified, with Groups A, B, C, and D consisting of 35, 90, 98, and 29 individuals, respectively.

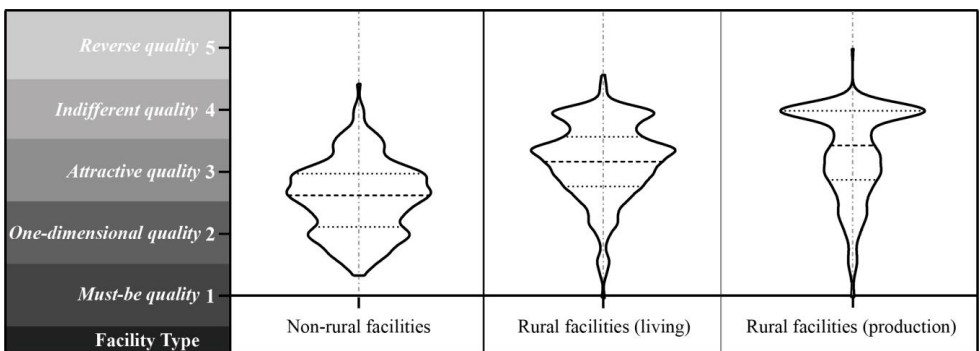

**Figure 7.** Violin plots of the mean values of the individual residents' cognitive attributes.

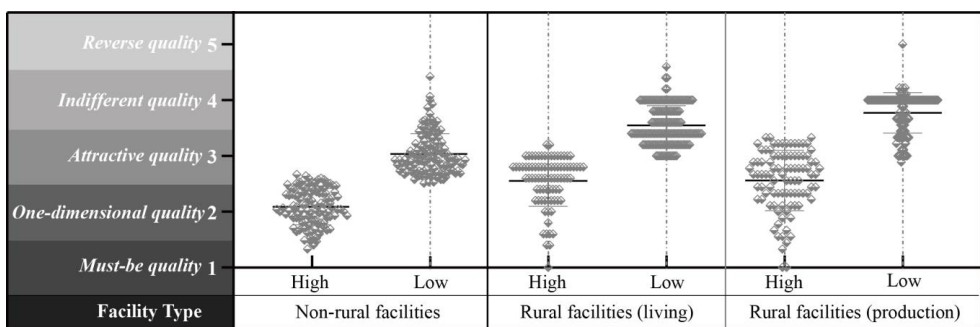

**Figure 8.** Distribution of the mean values of the individual residents' cognitive attributes after clustering.

*3.2. Demand Characteristics of Residents in Different Urbanization Stages*

After determining the individual residents' cognitive attributes regarding facilities, a further assessment of the needs of the groups was necessary. In the traditional Kano model, when there are multiple cognitive attributes that are close to or have a percentage similar to the highest-rated attribute, defining the attribute category based on the maximum value may result in some people's needs being ignored [57]. This study applied better–worse coefficient analysis to identify the similarity of individuals' cognitive attributes. Better and worse values represent the degrees of improvement in the satisfaction of residents after a facility is provided and those of the increase in dissatisfaction after it is not provided, respectively [58]. The calculation method for better and worse values is as follows:

$$\text{Better (SI)} = (A + O)/(A + O + M + I) \text{ and} \tag{1}$$

$$\text{Worse (DSI)} = -1 \times (O + M)/(A + O + M + I), \tag{2}$$

The better and worse values were used to plot a four-quadrant diagram based on their absolute values. As better and worse values are continuous, the intersection point of the axes was established as the average value. The four quadrants represent the following categories: one-dimensional (I), attractive (II), indifferent (III), and must-be (IV) qualities. Consequently, the four groups' facility attributes were classified (Table 5, Figures 9–12).

Upon examining the number of cognitive attributes for each of the four groups (Figure 13), it was observed that group C displayed the weakest sense of urgency for facilities, with only four must-be quality facilities (6.78%) and 24 indifferent quality facilities (40.68%). Group A followed Group C in rank, with nine must-be quality facilities (15.25%). In contrast, Group B displayed the strongest sense of urgency for facilities, with a larger number of must-be and one-dimensional quality facilities; it was observed that 30 facilities (50.85%) could cause dissatisfaction if they were not planned. Furthermore, the study ranked the urgency levels of cognitive attributes for each facility across the four groups. For instance, with regard to a clinic (A = 1, B = 1, C = 2, and D = 2), Groups A and B were classified as Rank 1, while Groups C and D were classified as Rank 3. The number of attributes with Ranks 1–4 for each group's perception of 59 facilities were measured (Figure 14). Group B perceived 52 facilities (88%) to correspond to Rank 1, which is sufficient for covering most of the needs of Groups A, C, and D. Upon further comparison with Groups A, C, and D, eight facilities showed a high level of urgency in ACD but were not covered by Group B (Figure 15). In terms of facility types, Groups A and B continued to have high demand for agriculture-related industry services. Residents in these groups considered agricultural materials and machine shops to be essential and placed great value in clan relationships formed by rural geography and familial ties. They also believed that cultural auditoriums were necessary. Group A had a strong demand for agriculture-related industry services and considered product sale sites and agricultural product acquisition stations to be indispensable. Meanwhile, Groups C and D were indifferent to whether rural facilities were available.

Table 5. Better–worse coefficient calculation and cognitive attribute results.

| Types | Facilities | Group A | | Group B | | Group C | | Group D | | Cognitive Attribute * | | | |
|---|---|---|---|---|---|---|---|---|---|---|---|---|---|
| | | Better | \|Worse\| | Better | \|Worse\| | Better | \|Worse\| | Better | \|Worse\| | A | B | C | D |
| Nonrural facilities | | | | | | | | | | | | | |
| Medical | Clinic | 0.4286 | 0.6571 | 0.2444 | 0.7444 | 0.4694 | 0.4184 | 0.3448 | 0.5517 | 1 | 1 | 2 | 2 |
| | Hospital | 0.6857 | 0.4000 | 0.3111 | 0.8111 | 0.5464 | 0.2990 | 0.1724 | 0.8276 | 2 | 2 | 2 | 1 |
| | Specialized clinic | 0.4412 | 0.1765 | 0.3889 | 0.3889 | 0.5052 | 0.1546 | 0.3793 | 0.2414 | 4 | 3 | 3 | 3 |
| | Emergency medical site (Nucleic-acid-testing site) | 0.4571 | 0.7143 | 0.1556 | 0.8778 | 0.5625 | 0.4583 | 0.3103 | 0.8276 | 2 | 1 | 2 | 2 |
| | Hugh sanatorium | 0.3939 | 0.1212 | 0.5444 | 0.2333 | 0.3918 | 0.0619 | 0.4138 | 0.1034 | 4 | 3 | 3 | 3 |
| Cultural and sports | Outdoor sports ground | 0.6765 | 0.2941 | 0.3556 | 0.5778 | 0.5104 | 0.1354 | 0.4483 | 0.4483 | 3 | 2 | 3 | 2 |
| | Indoor sports venue | 0.7429 | 0.0286 | 0.5667 | 0.3222 | 0.4896 | 0.0521 | 0.5517 | 0.1034 | 3 | 3 | 3 | 3 |
| | Cultural activity center | 0.6471 | 0.2647 | 0.4222 | 0.5111 | 0.5361 | 0.1546 | 0.4138 | 0.3448 | 3 | 3 | 3 | 3 |
| | Exhibition hall | 0.6286 | 0.0286 | 0.5111 | 0.1111 | 0.4639 | 0.0309 | 0.4483 | 0.1379 | 3 | 3 | 3 | 3 |
| Education | Kindergarten | 0.5429 | 0.4571 | 0.1000 | 0.9444 | 0.3196 | 0.4536 | 0.1034 | 0.9310 | 2 | 1 | 1 | 1 |
| | Primary school | 0.4571 | 0.5143 | 0.1111 | 0.9000 | 0.2887 | 0.4021 | 0.1034 | 0.9655 | 2 | 1 | 1 | 1 |
| | Junior school | 0.5143 | 0.4571 | 0.1889 | 0.8222 | 0.3711 | 0.3299 | 0.1034 | 0.9310 | 2 | 1 | 2 | 1 |
| | Senior high school | 0.5294 | 0.2353 | 0.3111 | 0.5000 | 0.3958 | 0.2292 | 0.3103 | 0.3793 | 3 | 3 | 2 | 3 |
| | Training institution | 0.3824 | 0.0588 | 0.5222 | 0.2667 | 0.3229 | 0.1250 | 0.2414 | 0.2069 | 4 | 3 | 4 | 3 |
| | Adult vocational and technical school | 0.2857 | 0.0286 | 0.4111 | 0.2333 | 0.2577 | 0.0515 | 0.2414 | 0.1379 | 4 | 3 | 4 | 3 |
| Social welfare | Home aged care service center | 0.6176 | 0.5294 | 0.2222 | 0.6444 | 0.4536 | 0.2268 | 0.3793 | 0.3103 | 2 | 1 | 2 | 3 |
| | Nursing home | 0.4857 | 0.2571 | 0.3000 | 0.4889 | 0.3608 | 0.1546 | 0.3793 | 0.2759 | 3 | 3 | 3 | 3 |
| | Service station for the disabled | 0.4412 | 0.2059 | 0.2556 | 0.4667 | 0.2784 | 0.1237 | 0.3103 | 0.2759 | 4 | 4 | 4 | 3 |
| | Social welfare institute | 0.5143 | 0.1143 | 0.3000 | 0.4444 | 0.3333 | 0.1042 | 0.3448 | 0.1379 | 3 | 3 | 4 | 3 |
| | Villager canteen | 0.5588 | 0.0882 | 0.4889 | 0.2556 | 0.4433 | 0.0722 | 0.3793 | 0.0690 | 3 | 3 | 3 | 3 |
| Administration | Neighborhood (village) service center | 0.3429 | 0.8857 | 0.1222 | 0.9444 | 0.3673 | 0.5000 | 0.2069 | 0.6552 | 1 | 1 | 2 | 1 |
| Commerce | Agricultural market | 0.6286 | 0.7429 | 0.1667 | 0.9889 | 0.5876 | 0.5670 | 0.1724 | 0.9655 | 2 | 1 | 2 | 1 |
| | Grocery | 0.6857 | 0.6000 | 0.1889 | 0.9444 | 0.5102 | 0.5306 | 0.2069 | 0.9310 | 2 | 1 | 2 | 1 |
| | Food store | 0.6857 | 0.3143 | 0.3111 | 0.7667 | 0.5567 | 0.2887 | 0.2069 | 0.7586 | 2 | 2 | 2 | 1 |
| | Pharmacy | 0.4571 | 0.6286 | 0.1111 | 0.9222 | 0.4694 | 0.4490 | 0.1379 | 0.8966 | 2 | 1 | 2 | 1 |
| | Bank | 0.4000 | 0.4000 | 0.1444 | 0.9667 | 0.4375 | 0.3750 | 0.1379 | 0.8966 | 1 | 1 | 2 | 1 |
| | Telecom business hall | 0.2571 | 0.1714 | 0.1444 | 0.8111 | 0.3542 | 0.2708 | 0.1034 | 0.6897 | 4 | 1 | 1 | 1 |
| | Beauty and hair salon | 0.6000 | 0.4571 | 0.1444 | 0.9111 | 0.4479 | 0.4375 | 0.1724 | 0.7931 | 2 | 1 | 2 | 1 |
| | Department store | 0.5429 | 0.1429 | 0.3333 | 0.6556 | 0.6392 | 0.2165 | 0.3103 | 0.6207 | 3 | 2 | 2 | 2 |
| | Cultural goods store | 0.3429 | 0.0857 | 0.2444 | 0.5889 | 0.3542 | 0.1354 | 0.0714 | 0.6071 | 4 | 1 | 4 | 1 |
| | Insurance institution | 0.2000 | 0.0000 | 0.1910 | 0.4270 | 0.1771 | 0.0833 | 0.0357 | 0.2500 | 4 | 4 | 4 | 4 |
| | Photographic studio | 0.3714 | 0.0857 | 0.2333 | 0.4222 | 0.1939 | 0.0714 | 0.0714 | 0.2500 | 4 | 4 | 4 | 4 |
| | Public bathroom | 0.2000 | 0.0000 | 0.2809 | 0.3146 | 0.1237 | 0.0619 | 0.1034 | 0.2414 | 4 | 3 | 4 | 4 |

**Table 5.** *Cont.*

| Types | Facilities | Group A Better | Group A \|Worse\| | Group B Better | Group B \|Worse\| | Group C Better | Group C \|Worse\| | Group D Better | Group D \|Worse\| | Cog. A | Cog. B | Cog. C | Cog. D |
|---|---|---|---|---|---|---|---|---|---|---|---|---|---|
| | Sales and maintenance department of non-motor vehicles | 0.4571 | 0.4571 | 0.1222 | 0.8444 | 0.2577 | 0.2165 | 0.1724 | 0.6207 | 2 | 1 | 1 | 1 |
| | Hardware store | 0.3143 | 0.2000 | 0.1667 | 0.8444 | 0.2268 | 0.1546 | 0.0690 | 0.7931 | 4 | 1 | 4 | 1 |
| | Express/postal service station | 0.4571 | 0.4286 | 0.1556 | 0.8889 | 0.4737 | 0.4632 | 0.2759 | 0.8621 | 2 | 1 | 2 | 2 |
| | Self-service cabinet | 0.4571 | 0.2571 | 0.4333 | 0.3778 | 0.4227 | 0.2268 | 0.2069 | 0.5517 | 3 | 3 | 2 | 1 |
| | Restaurant/teahouse | 0.5714 | 0.0857 | 0.3111 | 0.7444 | 0.4388 | 0.2959 | 0.3448 | 0.7241 | 3 | 2 | 2 | 2 |
| Infrastructure | Bus station | 0.4286 | 0.7714 | 0.1222 | 0.9444 | 0.4021 | 0.5670 | 0.1034 | 0.9655 | 1 | 1 | 2 | 1 |
| | Motor vehicle parking lot | 0.4571 | 0.5429 | 0.2000 | 0.8444 | 0.4694 | 0.5918 | 0.1724 | 0.8621 | 2 | 1 | 2 | 1 |
| | Public toilet | 0.3824 | 0.7059 | 0.1000 | 0.9778 | 0.3830 | 0.5213 | 0.1034 | 1.0000 | 1 | 1 | 2 | 1 |
| | Garbage collection point | 0.3030 | 0.8485 | 0.1111 | 1.0000 | 0.4157 | 0.6966 | 0.1034 | 1.0000 | 1 | 1 | 2 | 1 |
| Productive Service (Tourism, etc.) | Tourist reception center | 0.3714 | 0.0286 | 0.4205 | 0.1591 | 0.2577 | 0.0103 | 0.3448 | 0.0345 | 4 | 3 | 4 | 3 |
| | Tourist souvenir shop | 0.3235 | 0.0000 | 0.4659 | 0.0795 | 0.2474 | 0.0103 | 0.3103 | 0.0000 | 4 | 3 | 4 | 3 |
| | Hotel/Homestay | 0.2941 | 0.1765 | 0.1573 | 0.5730 | 0.1939 | 0.0714 | 0.0690 | 0.4138 | 4 | 1 | 4 | 4 |
| Productive service (Agriculture, etc.) | Agricultural machine shop | 0.3143 | 0.3143 | 0.0778 | 0.6667 | 0.1895 | 0.0526 | 0.1481 | 0.0370 | 1 | 1 | 4 | 4 |
| | Agricultural materials shop | 0.2571 | 0.6571 | 0.0667 | 0.7889 | 0.2128 | 0.1064 | 0.1481 | 0.2222 | 1 | 1 | 4 | 4 |
| | Grain-drying site | 0.2000 | 0.1143 | 0.1124 | 0.2022 | 0.1563 | 0.0104 | 0.1034 | 0.0345 | 4 | 4 | 4 | 4 |
| | Agricultural product storage station | 0.2353 | 0.1176 | 0.1910 | 0.1798 | 0.1340 | 0.0103 | 0.0690 | 0.0000 | 4 | 4 | 4 | 4 |
| | Product sales site | 0.4571 | 0.4000 | 0.2444 | 0.4444 | 0.2577 | 0.0000 | 0.1379 | 0.0690 | 2 | 4 | 4 | 4 |
| | Agricultural product acquisition station | 0.3429 | 0.4286 | 0.2778 | 0.3889 | 0.2371 | 0.0000 | 0.1034 | 0.0000 | 1 | 3 | 4 | 4 |
| | Farm tool storage station | 0.1250 | 0.0625 | 0.1744 | 0.1163 | 0.1809 | 0.0000 | 0.0690 | 0.0000 | 4 | 4 | 4 | 4 |
| | Agricultural technical education station | 0.5714 | 0.1143 | 0.4333 | 0.2889 | 0.2105 | 0.0000 | 0.1034 | 0.0345 | 3 | 3 | 4 | 4 |
| | Agricultural cooperative | 0.4571 | 0.2857 | 0.3371 | 0.2809 | 0.1354 | 0.0313 | 0.1034 | 0.0345 | 3 | 3 | 4 | 4 |
| Cultural and sports | Cultural auditorium | 0.5429 | 0.7143 | 0.3333 | 0.7333 | 0.4536 | 0.2062 | 0.2069 | 0.3103 | 2 | 2 | 3 | 4 |
| | Ancestral hall | 0.3714 | 0.1714 | 0.3483 | 0.2584 | 0.1563 | 0.0000 | 0.0357 | 0.0357 | 4 | 3 | 4 | 4 |
| | Temple | 0.4000 | 0.2000 | 0.3596 | 0.2360 | 0.2083 | 0.0208 | 0.0714 | 0.0357 | 4 | 3 | 4 | 4 |
| | Film-screening venue | 0.7714 | 0.2571 | 0.6111 | 0.2556 | 0.4896 | 0.0729 | 0.5517 | 0.1034 | 3 | 3 | 3 | 3 |
| | Broadcasting station | 0.3714 | 0.0571 | 0.2809 | 0.1798 | 0.2688 | 0.0108 | 0.2500 | 0.0000 | 4 | 3 | 4 | 3 |

(The leftmost "Types" column spans all rows as **Rural facilities**.)

* Cognitive Attribute: Must-be quality = 1, One-dimensional quality = 2, Attractive quality = 3, and Indifferent quality = 4.

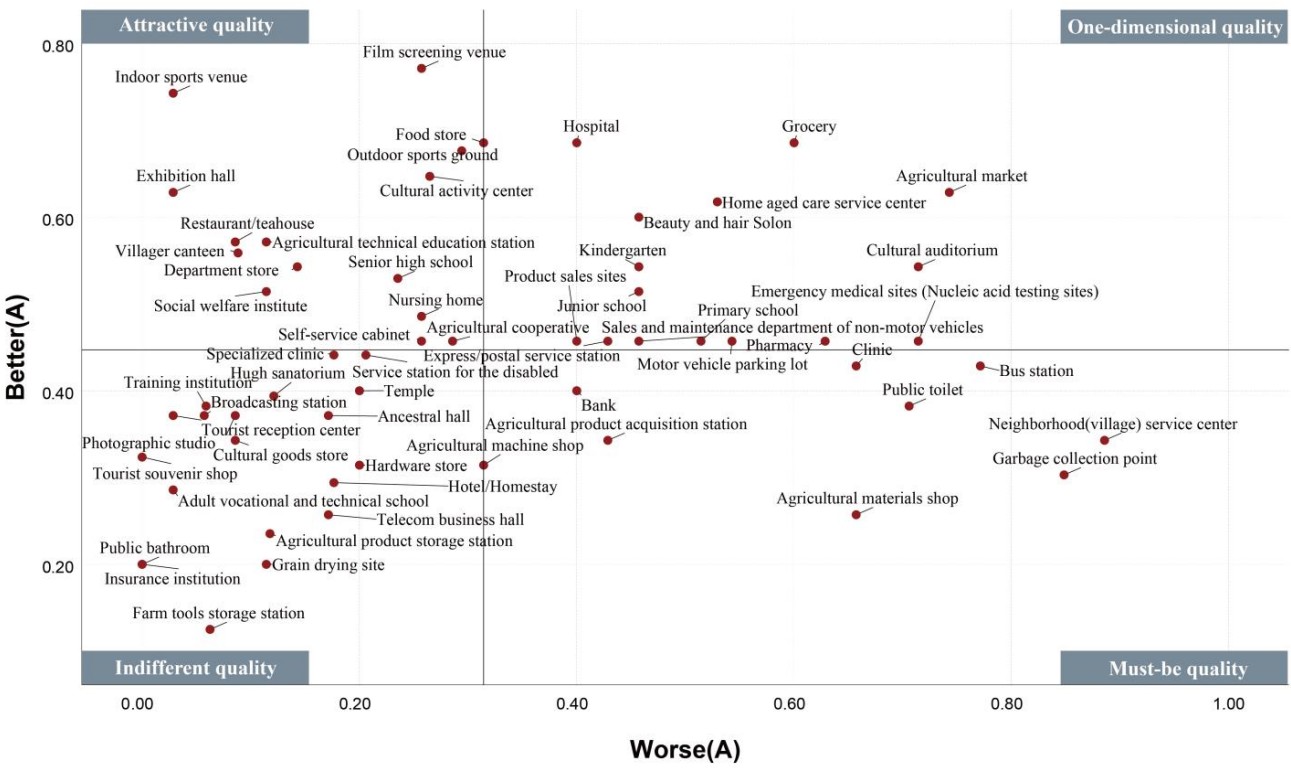

**Figure 9.** Classification of better–worse four-quadrant facility attributes (Group A).

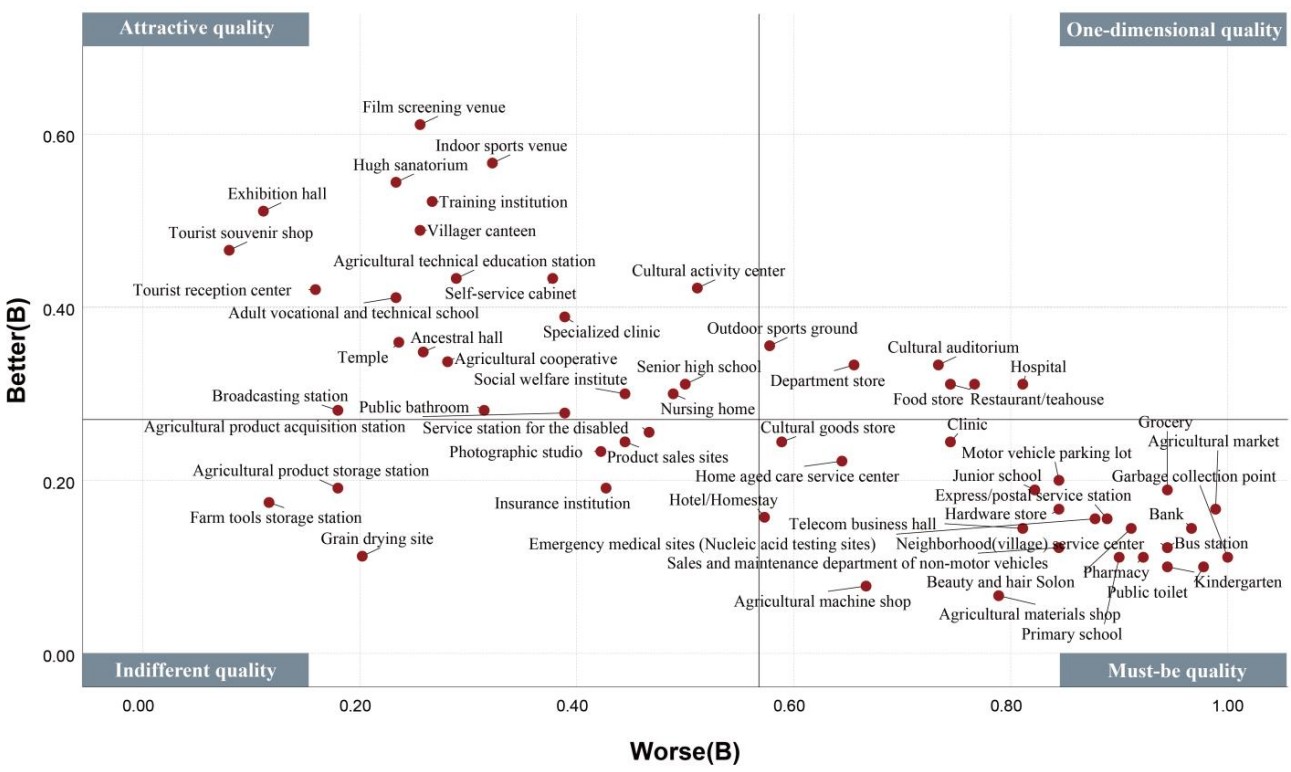

**Figure 10.** Classification of better–worse four-quadrant facility attributes (Group B).

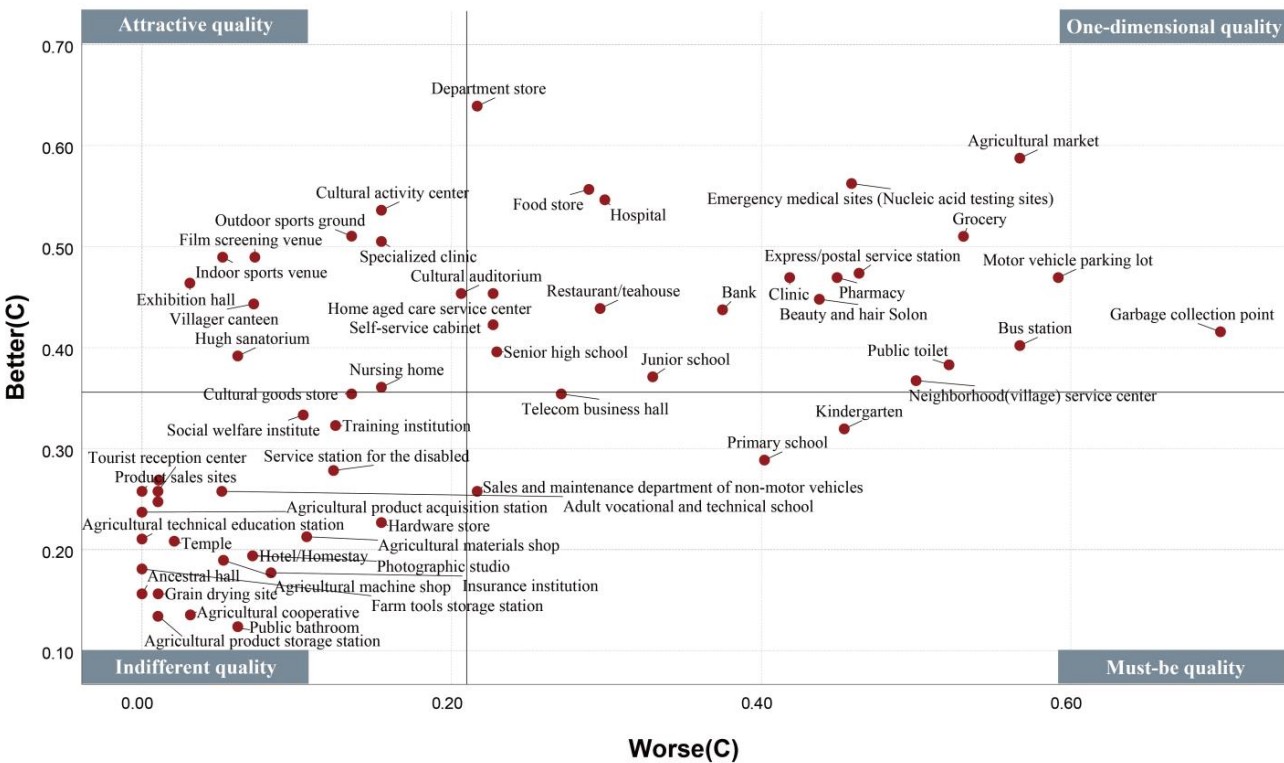

**Figure 11.** Classification of better–worse four-quadrant facility attributes (Group C).

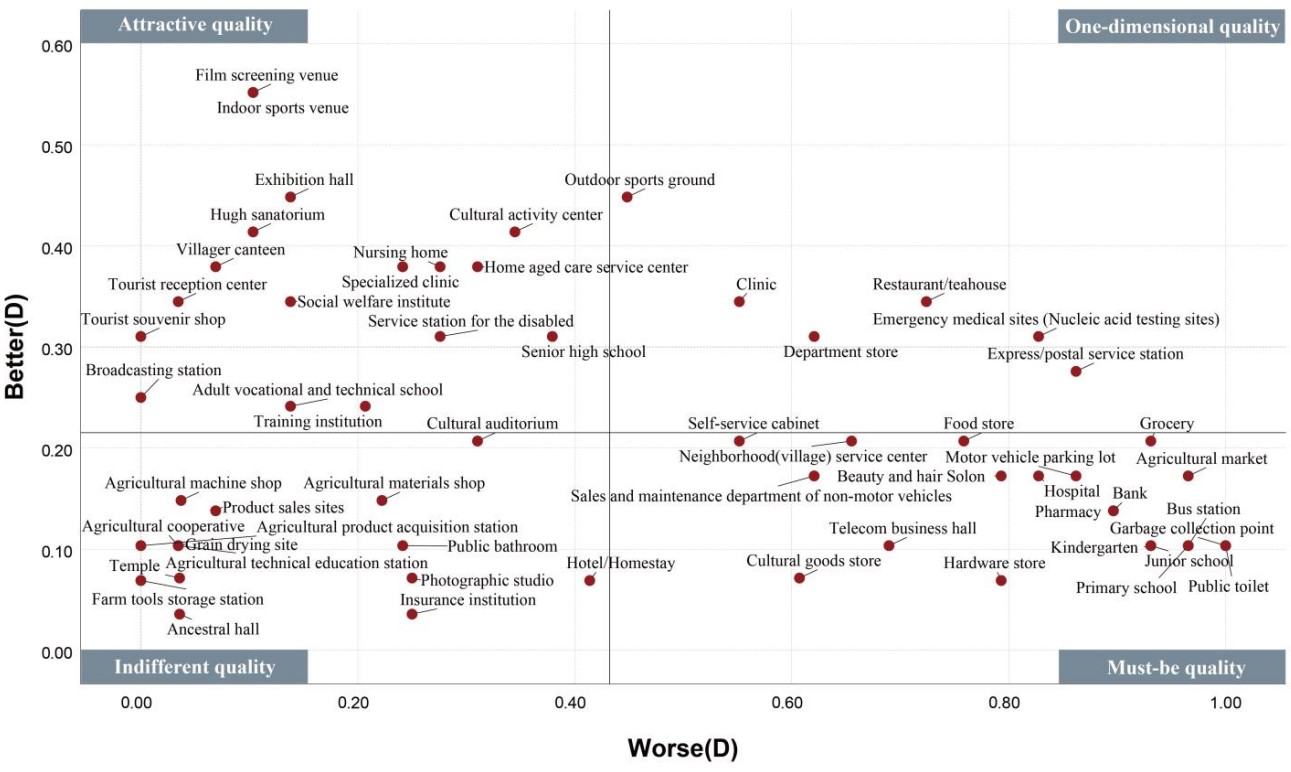

**Figure 12.** Classification of better–worse four-quadrant facility attributes (Group D).



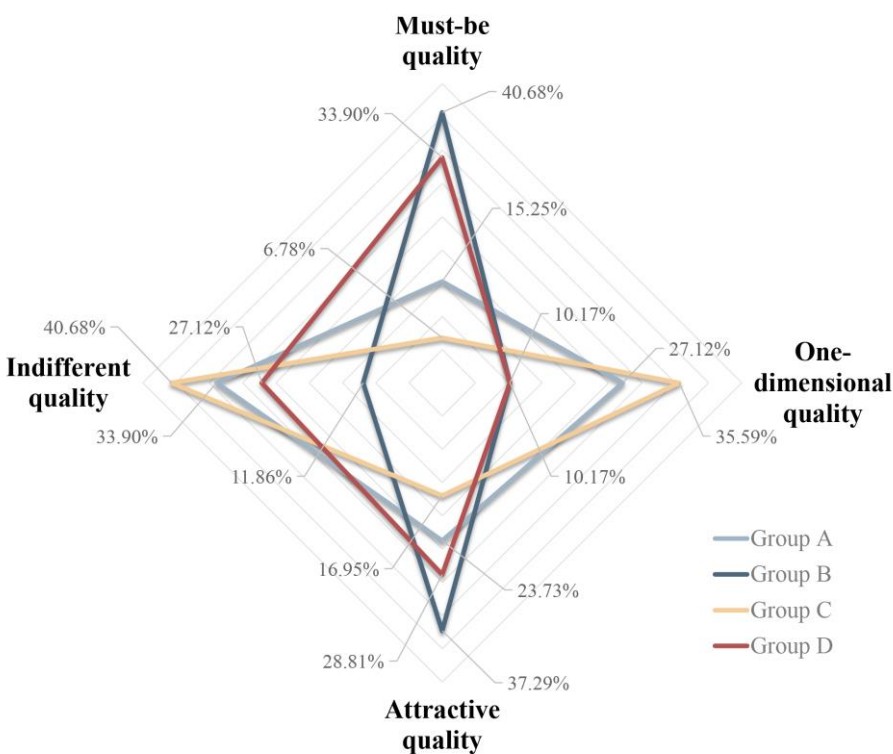

**Figure 13.** Radar chart of the number of facilities according to different cognitive attributes for the four groups.

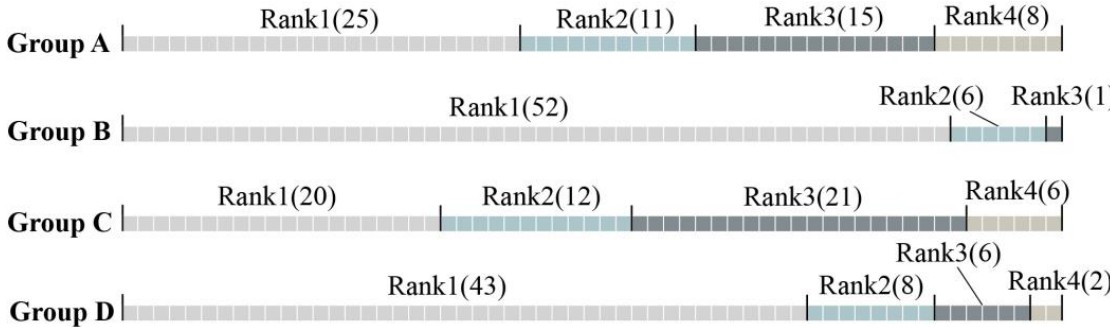

**Figure 14.** Statistics of the four groups' rankings of the urgency of each facility's cognitive attributes.

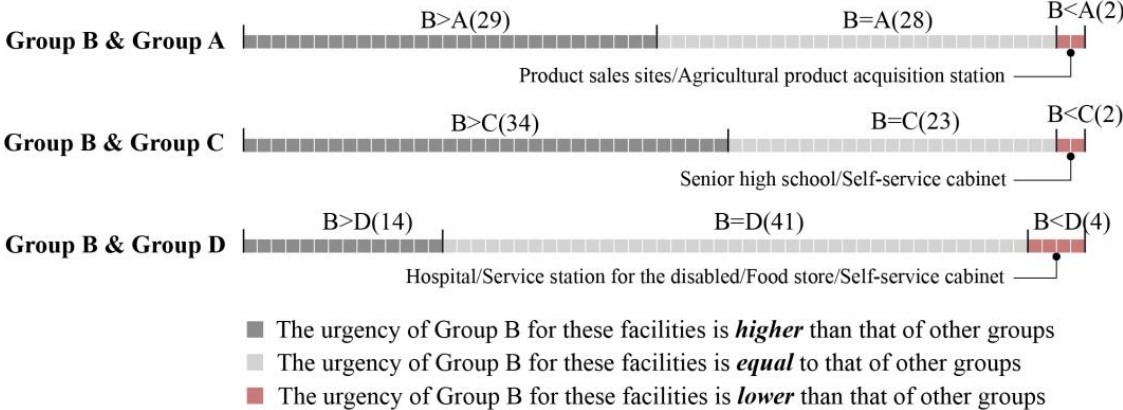

**Figure 15.** Comparative analysis chart of the urgency of Group B's needs versus the other three groups.

### 3.3. Matching Characteristics of the "Residential Area–Residential Facility" Relationship

According to the chi-square test (Table 6), there was a significant correlation between the individuals' urbanization levels and the type of residential area in which they lived ($p < 0.05$). Proportion analysis (Figure 16A) showed that two groups with lower levels of urbanization (Group A and Group B) tended to live in rural residential areas, while Group D tended to live in urban residential areas. The proportion of these types of residential areas in Group C was roughly equal. Due to the large population base of Group B, the proportion of residents in urban-biased residential areas could not be neglected (26.79%). According to a proportion analysis of the groups in different residential areas (Figure 17), the residents in Groups B, C, and D were the main residents in urban-biased residential areas (accounting for over 94.64% of the total), while those in Groups A, B and C were the main residents in rural-biased residential areas (accounting for over 97.14% of the total). This indicates a relatively matched relationship between the urbanization patterns for residents and residential areas.

**Table 6.** The chi-square test results regarding population attributes.

| Population Attributes | | Option | | Group A | Group B | Group C | Group D | *p* |
|---|---|---|---|---|---|---|---|---|
| (A) Residential Area Type | A1 | Urban-biased residential area | | 17.1% | 33.3% | 52.0% | 86.2% | 0.000 ** |
| | A2 | Rural-biased residential area | | 82.9% | 66.7% | 48.0% | 13.8% | |
| (B) Age Type | B1 | Youth (18–25) | | 2.9% | 3.3% | 9.2% | 3.4% | 0.015 * |
| | B2 | Middle-aged (25–65) | | 54.3% | 63.3% | 66.3% | 89.7% | |
| | B3 | Seniors (over 65) | | 42.9% | 33.3% | 24.5% | 6.9% | |
| (C) Registered Permanent Address | C1 | Rural | | 91.4% | 81.1% | 73.5% | 69.0% | 0.029 * |
| | C2 | Urban (experienced rural–urban migration) | | 8.6% | 6.7% | 4.1% | 3.4% | |
| | C3 | Urban | | 0 | 12.2% | 22.4% | 27.6% | |
| (D) Education Level | D1 | Did not complete primary School | | 11.4% | 3.3% | 12.2% | 0 | 0.028 * |
| | D2 | Completed primary school | | 25.7% | 25.3% | 17.3% | 13.8% | |
| | D3 | Completed junior school | | 14.3% | 26.7% | 19.4% | 34.5% | |
| | D4 | Completed senior high school | | 14.3% | 22.2% | 12.2% | 10.3% | |
| | D5 | Completed university and above | | 34.3% | 22.2% | 38.8% | 41.4% | |
| (E) Employment Status | Whether self-employed | - | No | 94.3% | 80.0% | 88.8% | 69.0% | 0.015 * |
| | | - | Yes | 5.7% | 20.0% | 11.2% | 31.0% | |
| | Whether farming | - | No | 80.0% | 75.6% | 88.8% | 100.0% | 0.006 ** |
| | | - | Yes | 20.0% | 24.4% | 11.2% | 0 | |

\* $p < 0.05$; \*\* $p < 0.01$.

The age, registered permanent residence, educational level, and employment status (farming or individual business) of the respondents showed a significant correlation with the residents' urbanization stage ($p < 0.05$). Through proportion analysis (Figure 16), it was gleaned that residents with a low level of urbanization tended to have rural household registration, be elderly, have lower levels of education, and primarily engage in agricultural activities. In contrast, residents with a high level of urbanization tended to have urban household registration, be middle-aged or young, have higher levels of education, engage in agricultural activities less frequently, and be self-employed.

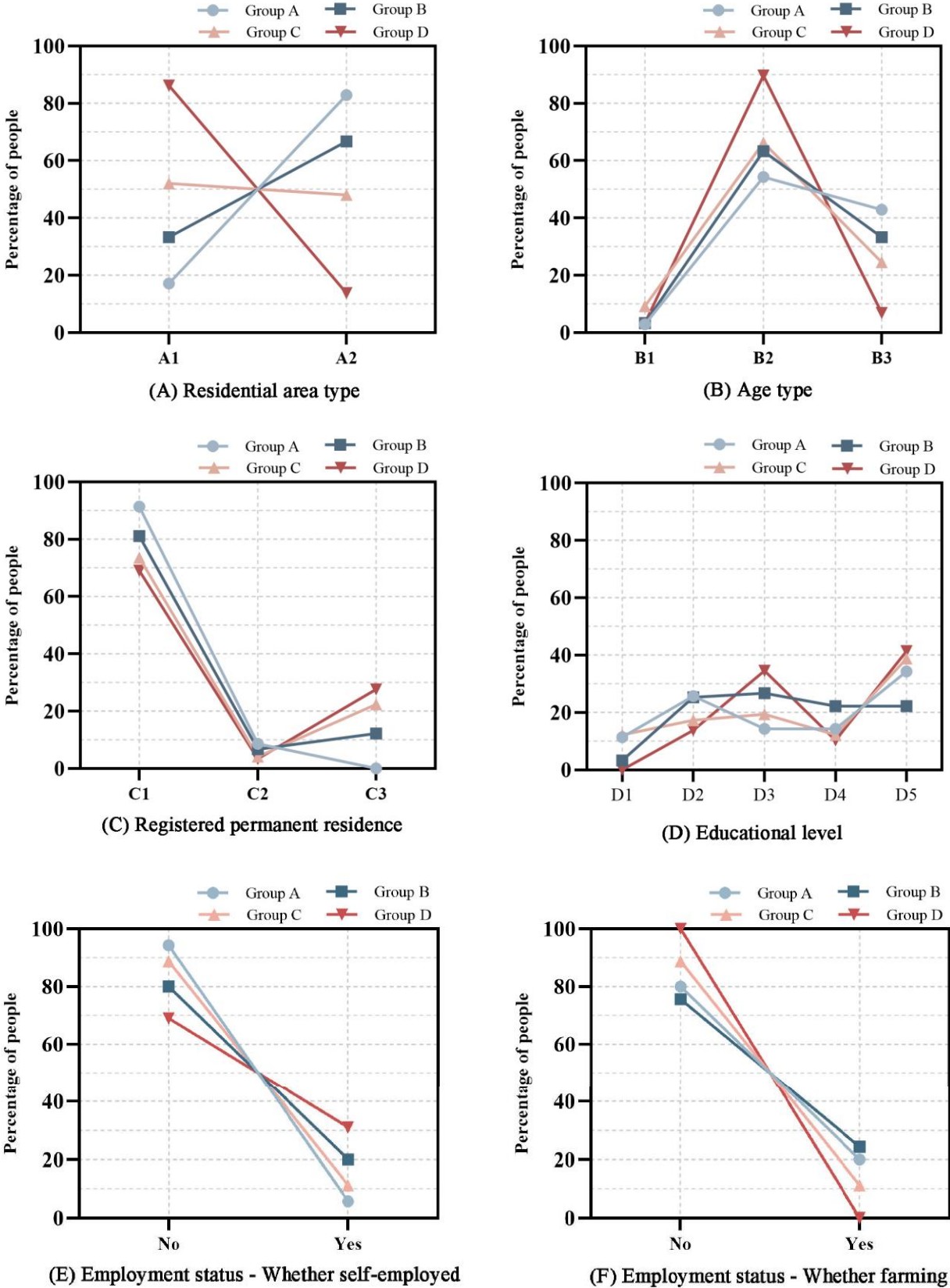

**Figure 16.** Proportion analysis of residents with different population attributes.

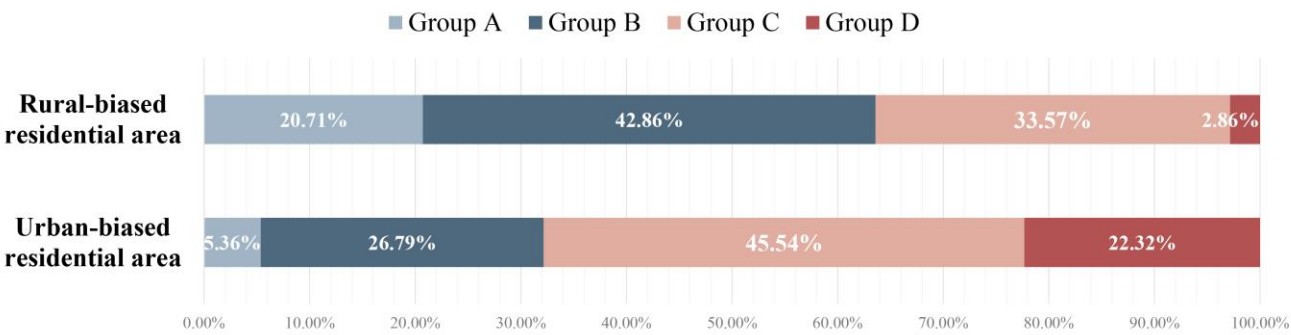

**Figure 17.** Percentage distribution of the four groups of residents in rural- and urban-biased residential areas.

## 4. Discussion

The above analysis clarifies the need characteristics of the different resident groups at different stages of urbanization. However, solely obtaining the demand characteristics of different urbanization stages is not sufficient for directly guiding practical applications. Therefore, the study further combines the needs of multiple groups based on the matching relationship of human–land urbanization patterns and proposes a guidance-control system for facility programming in rural- and urban-biased residential areas. Given the proportion of groups in these two residential areas, the populations of Groups B, C, and D accounted for the majority in urban-biased residential areas, while the populations of Groups A, B, and C accounted for the majority in rural-biased residential areas. To determine the specific levels of required or optional facilities, analytical principles must be developed to combine the three resident groups' cognitive attributes based on their need characteristics.

### 4.1. Guidance-Control Principle of Facility Programming Based on the Characteristics of Residents' Needs

The core objective of the "control" proposed by the guidance-control system is not to ensure the implementation of the best possible solution but rather the avoidance of the worst-case scenario [59]. Therefore, the goal of organizing "required items" is not to satisfy the needs of all groups simultaneously but to prioritize avoiding dissatisfaction from those with lower service expectations and ensure that their must-be and one-dimensional quality needs are met. "Optional items" consider the must-be and one-dimensional quality needs of residents with high service expectations as well as the attractive quality needs of all groups of residents, helping to prevent dissatisfaction and further enhance overall satisfaction.

Based on the results of the data analysis in Section 3, Group B has an equal or greater level of urgency for most facilities compared with the other three groups. As a group that requires consideration for rural- and urban-biased residential areas, Group B can cover the majority of the other groups' needs. However, for some facilities, Groups A, C, and D also showed a high level of urgency that was not covered by Group B. Therefore, Groups A, C, and D did not have absolutely low service expectations. In fact, each facility had two aspects with high and low levels of service expectations. In this study, the level of service expectations for each facility was screened, and the required and optional items were determined based on the needs of different groups with varying levels of service expectations. The establishment of the guidance-control system is based on the principle of progressively avoiding population dissatisfaction and progressively improving population satisfaction at each level. The specific steps are as follows.

Assume that the cognitive attributes of the three groups for n facilities are $X_{n1}$, $X_{n2}$, and $X_{n3}$ ($n \in [1, 59]$, $X \in [1, 4]$). Define the attributes of a facility recognized by a group with high service expectations using the following formula:

$$X_{ni} = \min [X_{n1}, X_{n2}, X_{n3}]. \tag{3}$$

The attributes of a facility are recognized by two groups with low service expectations according to the following formula:

$$X_{nj} = \max [X_{n1}, X_{n2}, X_{n3}] \text{ and} \tag{4}$$

$$X_{nk} = X_{n1} + X_{n2} + X_{n3} - X_{ni} - X_{nj}. \tag{5}$$

If a facility receives an unsatisfactory rating from residents with low service expectations because of its absence, it should be categorized as "Level 1: Required Items" (either $X_{nj}$ or $X_{nk}$ is one or two). The next level should meet the needs of the residents with high service expectations. Facilities that may result in dissatisfaction if not provided to this group should be designated as "Level 2: Optional Items 1" ($X_{ni}$ is one or two). After addressing dissatisfaction due to the absence of certain facilities among the different resident groups, the facilities that can further enhance satisfaction for the three groups of residents should be designated as "Level 3: Optional Items 2" ($X_{ni}$, $X_{nj}$, and $X_{nk}$ are all neither one nor two but three). In addition, there are types of facilities toward which all three groups have an indifferent attitude. These facilities transcend or fall behind the current production and living needs of residents and can thus be temporarily excluded from consideration. In this study, they are designated as "Level 4: Optional Items 3" ($X_{ni}$, $X_{nj}$, and $X_{nk}$ are all equal to four). The specific process is shown in Figure 18.

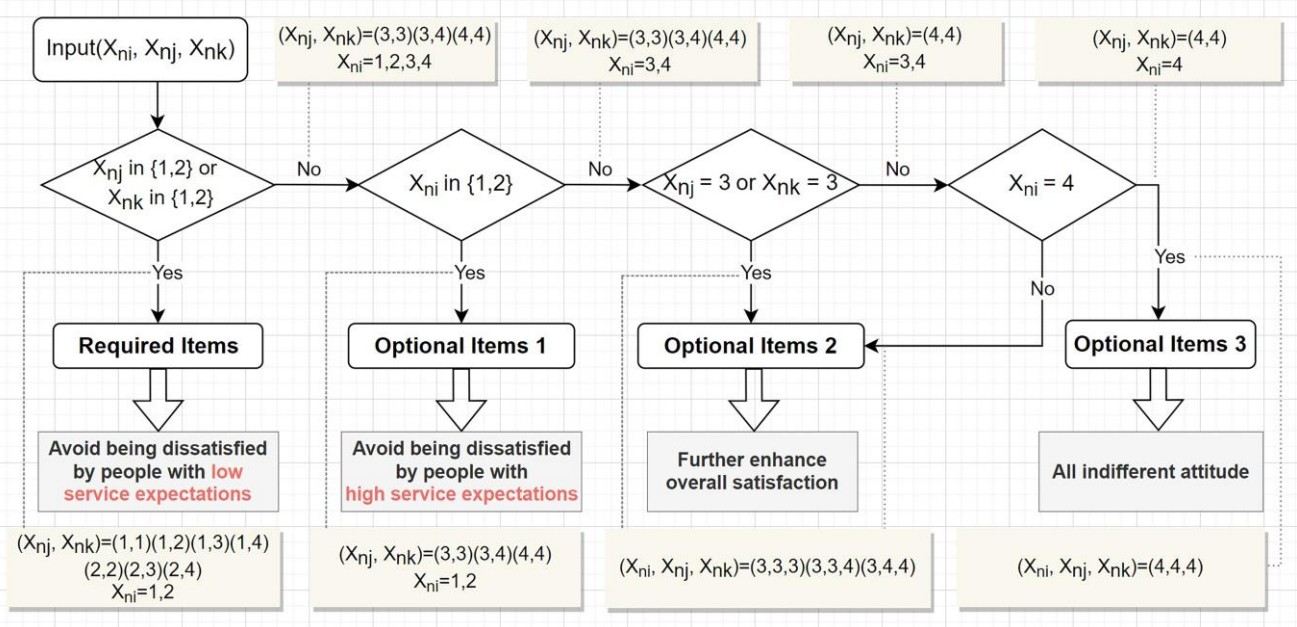

**Figure 18.** Flow chart of guidance-control principle.

In practical applications, the implementation of facility levels can be conducted according to the order of funding input, with required items being implemented first, followed by Optional Items 1 and 2. Required items refer to the facility levels that must be implemented. Optional Items 1 is recommended for implementation if economic conditions permit because such items can prevent dissatisfaction among all the groups. If funding conditions are sufficient, Optional Items 2 can be implemented.

### 4.2. Programming Guidance-Control System of Public Service Facilities in Small Towns Based on the Pattern of Human–Land Urbanization

In this section, based on the guidance-control principle, the classification and guidance of facility programming in rural and urban-biased residential areas are conducted, the order of facility programming is clarified, and a guidance-control system of "required + optional item" public service facilities is developed (Table 7).

**Table 7.** The "required + optional item" public service facility guidance-control system for rural- and urban-biased residential areas.

| | | Differentiation Items (Rural-Biased Residential Area) | Common Items | Differentiation Items (Urban-Biased Residential Area) |
|---|---|---|---|---|
| **The Relationship between Human–Land Urbanization Patterns** | | | **Land ≈ Human/Land (Urban-Biased Residential Area) > Human (Elderly, Low Education Level, Rural Household Registration, Engaged in Farming)** | |
| | | | **Land ≈ Human/Land (Urban-Biased Residential Area) < Human (Young, High Education Level, Urban Household Registration, Self-Employed)** | |
| Required Items | Medical | | Clinic, Hospital, Emergency medical site (Nucleic-acid-testing site) | |
| | Education | | Kindergarten, Primary school, Junior school | Senior high school |
| | Cultural and sports | Cultural auditorium | | Outdoor sports ground |
| | Social welfare | | Home aged care service center | |
| | Administration | | Neighborhood (village) service center | |
| | Commerce | | Agricultural market, Grocery, Food store, Pharmacy, Bank, Telecom business hall, Beauty and hair salon, Department store, Sales and maintenance department of non-motor vehicles, Express/postal service station, Restaurant/teahouse | Cultural goods store, Hardware store, Self-service cabinet |
| | Infrastructure | | Bus station, Motor vehicle parking lot, Public toilet, Garbage collection point | |
| | Productive Service | Agricultural machine shop, Agricultural materials shop | | |

**Table 7.** *Cont.*

| | | Differentiation Items (Rural-Biased Residential Area) | Common Items | Differentiation Items (Urban-Biased Residential Area) |
|---|---|---|---|---|
| The Relationship between Human–Land Urbanization Patterns | | **Land ≈ Human/Land (Urban-Biased Residential Area) > Human (Elderly, Low Education Level, Rural Household Registration, Engaged in Farming)** | | |
| | | **Land ≈ Human/Land (Urban-Biased Residential Area) < Human (Young, High Education Level, Urban Household Registration, Self-Employed)** | | |
| Optional Items 1 | Education | | Senior high school | |
| | Cultural and sports | Outdoor sports ground | | Cultural auditorium |
| | Commerce | Cultural goods store, Hardware store, Self-service cabinet | | |
| | Productive Service | Product sales site, Agricultural product acquisition station | Hotel/Homestay | Agricultural machine shop, Agricultural materials shop |
| Optional Items 2 | Medical | | Specialized clinic, Hugh sanatorium | |
| | Education | | Training institution, Adult vocational and technical school | |
| | Cultural and sports | | Indoor sports venue, Cultural activity center, Exhibition hall, Ancestral hall, Temple, Film-screening venue, Broadcasting station | |
| | Social welfare | | Nursing home, Social welfare institute | Service station for the disabled |
| | Commerce | | Villager canteen, Public bathroom | |
| | Productive Service | | Tourist reception center, Tourist souvenir shop, Agricultural technical education station, Agricultural cooperative | Agricultural product acquisition station |

**Table 7.** *Cont.*

| | | Differentiation Items (Rural-Biased Residential Area) | Common Items | Differentiation Items (Urban-Biased Residential Area) |
|---|---|---|---|---|
| **The Relationship between Human–Land Urbanization Patterns** | | **Land ≈ Human/Land (Urban-Biased Residential Area) > Human (Elderly, Low Education Level, Rural Household Registration, Engaged in Farming)** | | |
| | | **Land ≈ Human/Land (Urban-Biased Residential Area) < Human (Young, High Education Level, Urban Household Registration, Self-Employed)** | | |
| Optional Items 3 | Social welfare | Service station for the disabled | | |
| | Commerce | | Insurance institution, Photographic studio | |
| | Productive Service | | Grain-drying site, Agricultural product storage station, Farm tool storage station | Product sales site |

In the practical programming of public service facilities in small towns, it is important to note that not all residential areas have matching levels of human–land urbanization. Therefore, the first step should be to assess the degree of urbanization for the residential areas and their residents, based on which an appropriate programming scheme can be selected. There are three possibilities in the matching relationship of human–land urbanization: first, a high degree of matching between the urbanization of residential areas and residents (land ≈ human), as is the case in the sample in this study; second, the residential area's urbanization is occurring faster than the residents' urbanization (land > human), which is more common in apartment resettlement areas; and third, the residential area's urbanization is lagging behind the level of residents' urbanization (land < human), which is more common in economically developed rural areas. Thus, for urban-biased residential areas whose urbanization is faster than that of their residents, facilities should be constructed according to the program of rural residential areas, as the residents tend to be elderly, have lower levels of education and rural household registration, and still be engaged in farming. Meanwhile, for rural-biased residential areas whose urbanization lags behind that of their residents, facilities should be constructed according to the needs of urban residential areas, as the residents tend to be younger, possess higher levels of education, generally have urban household registration, and be engaged in self-employment activities.

Unlike previous studies, this study employed the Kano model as a methodological approach to identify facility-related cognitive attributes among individual residents and different urbanization-stage groups. In previous surveys conducted using the Likert scale method, it was observed that individuals with a high level of service expectation generally displayed a positive attitude towards most items. However, it is not necessarily the case that all items exhibiting a positive attitude should be given the highest priority for these individuals. When using the negative questions of the Kano model, it was found that interviewees actually responded with "I can live with it if it is not satisfied" instead of "dissatisfied" for certain positive attitude items. Conversely, for individuals with low service expectations, although they generally exhibited a negative attitude towards most items, some items were still considered "dissatisfied" when the negative question of the Kano model was asked. Therefore, the Kano model can mitigate the bias in survey results caused by overly positive or negative attitudes, effectively reflecting objective facts and accurately identifying cognitive attributes. This holds substantial significance when establishing the guidance-control hierarchy of public service facility programming. The rational selection of methods ensures that the results have certain distinctions compared to other studies. While some studies generally qualitatively propose a "guidance-oriented" approach to the programming of public service facilities in small towns, such as preserving certain rural-specific facilities for residents undergoing nearby urbanization [32], this study systematically categorized rural and nonrural facility types. It quantitatively determined the required or optional levels of different facility types based on the needs of residents at different stages of urbanization. As a result, this study provides directly applicable guidance and control strategies.

## 5. Conclusions

As an intermediary link in the "city–town–village" system, small towns serve as crucial support for the promotion of nearby urbanization. Compared with "remote urbanization", which entails the relocation of a rural population to a distant urban area, "nearby urbanization" alleviates the problem of "hollow villages" in rural areas caused by labor outflows. Additionally, nearby urbanization helps enhance rural residents' rights to public services and establish a connection to social security. However, nearby urbanization has resulted in small towns having complex urbanization patterns that differ greatly from those of cities and rural areas. This makes it difficult to determine the programming of public service facilities using rigid indicators. Therefore, this study introduced the concept of "guidance and control", proposing a suitable programming method that incorporates rigid

control and a certain degree of flexibility based on residents' needs, aiming to overturn the current "one-size-fits-all" facility allocation approach.

This study suggests that the level of need for rural and nonrural facilities reflects the different stages of residents' urbanization. Therefore, rural and nonrural facility types were distinguished, and a "rural and nonrural" facility need coupling model was constructed to classify residents into four different urbanization stages. The need systems and population attribute characteristics of each group were analyzed. Furthermore, this study considered the feasibility of implementing its results in other small towns to avoid the complex steps required for decision-makers to identify urbanization types and proportions of residents. Thus, this study not only explored the needs of different stages of urbanization but also deduced the "residential area–resident facility" matching relationship. The study clarified the analyzed residents' urbanization characteristics under the patterns of rural- and urban-biased residential areas and summarized the programming guidance-control system for public service facilities within this relationship.

This study also has certain limitations. Firstly, as a pioneering study on the suitability of public service facilities programming in small towns, this study divided residential areas into two categories based on their degrees of urbanization, namely, rural- and urban-biased areas. However, it did not address specific types of residential areas. Future studies may further explore the relationship between the human–land urbanization characteristics of various types of residential areas and develop a more comprehensive and refined facility programming system. Secondly, the selected sample in this study exhibits a matched urbanization pattern between human and land. Based on the results, we derived guidance-control schemes for "required + optional" public service facility provisions under two mismatched scenarios: "land > human" and "land < human". Future studies should consider selecting samples that represent other human–land urbanization pattern mismatches to further validate and refine the guidance-control system. Thirdly, this study focused on the functional aspect of facility programming, addressing the question of "what to construct." However, a practical application process also involves the implementation of spatial aspects, addressing the question of "where to construct", which requires the cross-functional integration of the different types of residential areas and the determination of accessibility and the corresponding life cycle of facilities. Therefore, the programming path for the suitability of public service facilities in small towns requires further exploration. While this study examined only one aspect of facility programming, it offers valuable insights for local governments and planning and construction departments with respect to developing and revising technical specifications, standards, and guidelines and updating public service facilities through a combination of rigid and flexible approaches.

**Author Contributions:** Conceptualization, Z.Q. and Y.W.; methodology, Z.Q. and Y.W.; software, Z.Q. and Y.W.; validation, Z.W. and Y.Z.; formal analysis, Z.Q. and Y.W.; resources, Z.Q. and Y.W.; data curation, Z.W. and Y.Z.; investigation, Z.Q., Y.W., J.W., Z.W. and Y.Z.; writing—original draft preparation, Z.Q., Y.W. and J.W.; writing—review and editing, Z.Q., Y.W. and J.W.; visualization, Y.W.; supervision, Y.Z.; project administration, Z.Q. and Y.Z.; funding acquisition, Z.Q. All authors have read and agreed to the published version of the manuscript.

**Funding:** This research was funded by the National Natural Science Foundation of China (52278044).

**Data Availability Statement:** Data supporting reported results can be found at https://pan.zju.edu.cn/share/d6f0d6c31aede925dec00c2567 (accessed on 30 April 2023).

**Acknowledgments:** The authors would like to express their sincere gratitude to the support received from the students of Zhejiang University in collecting the data, and the assistance in the research work provided by the staff of the S Town government.

**Conflicts of Interest:** The authors have no conflict of interest to declare.

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
