# Peer review of "Needs Hierarchy for Public Service Facilities and Guidance-Control Programming in Small Chinese Towns Influenced by Complex Urbanization of Residents: The Evidence from Zhejiang"

_land, doi:10.3390/land12061205_

Round 1
Reviewer 1 Report
"Needs Hierarchy for Public Service Facilities and Guidance-Control Programming in Small Chinese Towns Caused by Complex Urbanization of Residents"
The subject of the article covers the problems of urbanization in the areas of small towns in terms of their development, taking into account the needs of residents, with particular emphasis on the increase in the quality of public services.
The content of the manuscript is of high quality, although it seems that the authors unnecessarily tried to include two topics in the content of the research - closely related, but could also be included in two separate articles.
The abstract in the initial part introduces the subject of the research a bit too long, while in the final part there is too little information about the research results obtained.
The research methodology was presented in great detail, and the results were presented in a very comprehensive and legible, although sometimes quite complicated, way.
The problem starts when we get to the discussion. Presented in 4.1 ("Guidance-Control Principle of Facility Programming Based on the Characteristics of 406 Residents' Needs") and 4.2 ("Programming Guidance-Control System of Public Service Facilities in Small Towns Based on the Pattern of Human–Land Urbanization") is in practice the second, separate part of the research. This also applies to the figure 18 included in the discussion and the content of table 3. Does the content of these subsections correspond to what should be the discussion of the results presented in the previous chapter? I have serious doubts about the current shape of this chapter - the authors need to think about how to solve this problem.
Other elements of research at a good level.
Author Response
Dear reviewer:
Many thanks for your comments. The manuscript has been revised accordingly. The details are listed as below:
Point 1: It seems that the authors unnecessarily tried to include two topics in the content of the research - closely related, but could also be included in two separate articles.
Response 1: As expert has pointed out, the manuscript indeed "includes two topics in the content of the research - closely related, but could also be included in two separate articles." One part reveals the matching mechanism of "residential area-resident-facility," while the other part is a derived guidance-control system based on this mechanism to address the issue of mismatched supply and demand of public service facilities. Actually, both parts of the content can indeed be independent research topics. However, we hope to present the process of applying and solving problems based on mechanisms derived from objective facts, which is an attempt to transform the findings of hermeneutic research into practical research. Additionally, we did consider splitting the content into two separate articles during the writing process, but we were concerned about the continuity and scientific rigor of the second topic. Therefore, we greatly appreciate the valuable advice from the expert, which provides significant guidance for us in managing the volume of the journal paper and research.
Point 2: The abstract in the initial part introduces the subject of the research a bit too long, while in the final part there is too little information about the research results obtained.
Response 2: We have added comparisons with other studies in the Discussion chapter in terms of methods and results:"In comparison to previous studies, this study employed the Kano model as a methodological approach to identify facility cognitive attributes among individual residents and different urbanization stage groups. In previous surveys conducted using the Likert scale method, it was observed that individuals with a high level of service expectation generally displayed a positive attitude towards most items. However, it is not necessarily the case that all items exhibiting a positive attitude should be given the highest priority for these individuals. When using the negative questions of the Kano model, it was found that interviewees actually responded with 'I can live with it if it is not satisfied' instead of 'dissatisfied' for certain positive attitude items. Conversely, for individuals with low service expectation, although they generally exhibited a negative attitude towards most items, some items were still considered 'dissatisfied' when the negative question of the Kano model is asked. Therefore, the Kano model can mitigate the bias in survey results caused by overly positive or negative attitudes, effectively reflecting objective facts and accurately identifying cognitive attributes. This holds substantial significance when establishing the guidance-control hierarchy of public service facility programming. The rational selection of methods ensures that the results have certain distinctions compared to other studies. While some studies generally qualitatively propose a 'guidance-oriented' approach to the programming of public service facilities in small towns, such as preserving certain rural-specific facilities for residents undergoing nearby urbanization [32]. This study, in contrast, systematically categorizes rural and nonrural facility types. It quantitatively determines the required or optional levels of different facility types based on the needs of residents at different stages of urbanization. As a result, this study provides directly applicable guidance and control strategies."
Point 3: The problem starts when we get to the discussion. Presented in 4.1 ("Guidance-Control Principle of Facility Programming Based on the Characteristics of 406 Residents' Needs") and 4.2 ("Programming Guidance-Control System of Public Service Facilities in Small Towns Based on the Pattern of Human–Land Urbanization") is in practice the second, separate part of the research. This also applies to the figure 18 included in the discussion and the content of table 3. Does the content of these subsections correspond to what should be the discussion of the results presented in the previous chapter? I have serious doubts about the current shape of this chapter - the authors need to think about how to solve this problem.
Response 3: We have optimized the research flow chart in Figure 4 to clarify the relationship between Chapter 3 and Chapter 4. The steps have been correspondingly aligned with the specific sections of two chapters. Chapter 3 identifies "the needs of people at different stages of urbanization" (3.1 & 3.2). However, merely obtaining the needs of people at different stages of urbanization is not sufficient, as it does not directly guide practical implementation. Ultimately, it is essential to shift the focus towards the residential area and develop a programming system specifically tailored to different residential areas. Therefore, in section 3.3, the manuscript analyzes the "composition" and "population attribute characteristics" of different urbanization stages among residents in different residential areas. In section 4.1, based on the "composition" (3.3), a principle is constructed to integrate the needs of three types of residents in rural-biased and urban-biased residential areas. In section 4.2, "the needs of people at different stages of urbanization" (3.1 & 3.2) is input into the principle, and a guidance-control system is derived based on the "population attribute characteristics" (3.3) under different urbanization patterns.
Figure 4. Research flow chart
We have revised the relevant descriptions of Figure 18 and Table 3 in the manuscript.
Regarding Figure 18, we have added a description of the fundamental principles of guidance and control: "The establishment of the guidance-control system is based on the principle of progressively avoiding population dissatisfaction and progressively improving population satisfaction at each level." This is the basis of the process design in Figure 18.
Regarding Table 3, we have explained the meanings and functions of "Dysfunctional" and "Functional" in section 2.2.2:“It suggests that not every service provision or nonprovision will respectively increase or decrease users’ satisfaction with the service. There are typically two types of satisfaction assessment questions for users: positive and negative. Positive refers to the satisfaction of the respondent when the service attribute is present or functioning properly (Functional). Negative refers to the satisfaction of the respondent when the service attribute is lacking or experiencing malfunction (Dysfunctional) [54]. The responses to positive and negative questions are correlated, reflecting five attributes: must-be (M), one-dimensional (O), attractive (A), indifferent (I), and reverse quality (R) (Figure 3). The Kano evaluation table (Table 3) maps the responses to positive and negative questions to cognitive attributes. " And provide an example to illustrate the usage of this table:"For example, if a respondent chooses 'Dissatisfied' for dysfunctional, and 'It should be that way,''I am indifferent,' or 'I can live with it' for functional, the corresponding cognitive attribute in the Kano evaluation table is the Must-be quality attribute (M)."
The above has been revised the responses of the manuscript. Your careful review has provided more help to our study clearer and more comprehensive. Thanks again for the advice of expert!
Reviewer 2 Report
The paper is consistent with MDPI - Land and fits in the overall journal scope.
The article is well structured. The article presents the results of the research according to the set methodology.
Table 3 is more difficult to read, and it is suggested to consider the possibilities of its improvement.
In the Discussion, besides the comparison of the results with the other authors' results, it would be desirable to comment on the differences in methodological approach in relation to other authors who dealt with the same issue.
The authors should also highlight current limitations of their study.
Author Response
Dear reviewer:
Many thanks for your comments. The manuscript has been revised accordingly. The details are listed as below:
Point 1: Table 3 is more difficult to read, and it is suggested to consider the possibilities of its improvement.
Response 1: Regarding Table 3, we have provided additional explanations of the meanings and implications of "Dysfunctional" and "Functional" in section 2.2.2:
"It suggests that not every service provision or nonprovision will respectively increase or decrease users' satisfaction with the service. There are typically two types of satisfaction assessment questions for users: positive and negative. Positive refers to the satisfaction of the respondent when the service attribute is present or functioning properly (Functional). Negative refers to the satisfaction of the respondent when the service attribute is lacking or experiencing malfunction (Dysfunctional) [54]. The responses to positive and negative questions are correlated, reflecting five attributes: must-be (M), one-dimensional (O), attractive (A), indifferent (I), and reverse quality (R) (Figure 3). The Kano evaluation table (Table 3) maps the responses to positive and negative questions to cognitive attributes." And provide an example to illustrate the usage of this table:" For example, if a respondent chooses 'Dissatisfied' for dysfunctional, and 'It should be that way,''I am indifferent,' or 'I can live with it' for functional, the corresponding cognitive attribute in the Kano evaluation table is the Must-be quality attribute (M)."
Point 2: In the Discussion, besides the comparison of the results with the other authors' results, it would be desirable to comment on the differences in methodological approach in relation to other authors who dealt with the same issue.
Response 2: We have added comparisons with other studies in the Discussion section in terms of methods and results.
During the course of the study, we also conducted comparisons among different methods and ultimately selected the application of the Kano model to identify facility cognitive attributes among individual residents and different urbanization stage groups. "In previous surveys conducted using the Likert scale method, it was observed that individuals with a high level of service expectation generally displayed a positive attitude towards most items. However, it is not necessarily the case that all items exhibiting a positive attitude should be given the highest priority for these individuals. When using the negative questions of the Kano model, it was found that interviewees actually responded with 'I can live with it if it is not satisfied' instead of 'dissatisfied' for certain positive attitude items. Conversely, for individuals with low service expectation, although they generally exhibited a negative attitude towards most items, some items were still considered 'dissatisfied' when the negative question of the Kano model is asked. Therefore, the Kano model can mitigate the bias in survey results caused by overly positive or negative attitudes, effectively reflecting objective facts and accurately identifying cognitive attributes. This holds substantial significance when establishing the guidance-control hierarchy of public service facility programming."
The rational selection of methods ensures that the results have certain distinctions compared to other studies. In terms of the results, "some studies generally qualitatively propose a "guidance-oriented" approach to the programming of public service facilities in small towns, such as preserving certain rural-specific facilities for residents undergoing nearby urbanization [32]. This study, in contrast, systematically categorizes rural and nonrural facility types. It quantitatively determines the required or optional levels of different facility types based on the needs of residents at different stages of urbanization. As a result, this study provides directly applicable guidance and control strategies".
Point 3: The authors should also highlight current limitations of their study.
Response 3: The limitations of this manuscript in the conclusions chapter really need to be consolidated. We summarized three limitations and pointed out the corresponding further directions:
"This study also has certain limitations. Firstly, as a pioneering study on the suitability of public service facilities programming in small towns, this study divided residential areas into two categories based on their degree of urbanization; namely, rural- and urban-biased areas. However, it did not address specific types of residential area. Future studies may further explore the relationship between human–land urbanization characteristics of various types of residential areas and form a more comprehensive and refined facility programming systems (limitation 1). Secondly, the selected sample in this study exhibits a matched urbanization pattern between human and land. Based on the results, it derives guidance-control schemes for 'required + optional' public service facility provisions under two mismatched scenarios: 'land > human' and 'land < human'. Future study should consider selecting samples that represent other human-land urbanization pattern mismatches to further validate and refine the guidance-control system (limitation 2). Thirdly, this study focused on the functional aspect of facility programming, addressing the question of 'what to construct.' However, a practical application process also involves the implementation of spatial aspects, addressing the question of 'where to construct,' which requires cross-functional integration of the different types of residential areas and the determination of accessibility, as well as the corresponding life cycle of facilities (limitation 3). Therefore, the programming path for the suitability of public service facilities in small towns requires further exploration."
The above has been revised the responses of the manuscript. Your careful review has provided more help to our study clearer and more comprehensive. Thanks again for the advice of expert!
Reviewer 3 Report
In general, this manuscript is well-written and the research presented is timely and interesting. The reviewer has some minor comments as follows:
1, The method part should give a detailed description of the questionnaire and survey process, such as what questions are used to measure each latent variable and how to ensure the representativeness and effectiveness of samples in the research process.
2, The author(s) are recommended to change the subtitle of the article to the style of evidence from Zhejiang.
3, The author(s) are recommended to describe their limitations and further directions separately.
Minor editing of English language required
Author Response
Dear reviewer:
Many thanks for your comments. The manuscript has been revised accordingly. The details are listed as below:
Point 1: The method part should give a detailed description of the questionnaire and survey process, such as what questions are used to measure each latent variable and how to ensure the representativeness and effectiveness of samples in the research process.
Response 1: The description of the questionnaire and investigation process in the Methods section of this manuscript is indeed unclear.
We have changed the heading of section 2.3.2 to "Questionnaire Design and Investigation Process" and provided a detailed description of the components of the questionnaire. "The survey questionnaire comprises two sections: gathering basic demographic information about residents and conducting a satisfaction survey using the Kano model. The basic demographic information serves to evaluate population attributes across various stages of urbanization. It encompasses details such as residential area, gender, age, registered permanent address, educational level, religious beliefs, household responsibilities, employment status, annual household income, marital status, and living arrangements with family members, among others. The satisfaction survey specifically targets each facility and incorporates a combination of positive and negative questions. " and provide an "example questionnaire (Table 4)" based on the Kano model.
Table 4. Example of satisfaction questionnaire based on the Kano model.
We have also provided a detailed description of the investigation process. "Before conducting the investigation, it is important to provide training to the investigators, simulate investigate scenarios, which mitigates potential information col-lection errors resulting from subjective interpretation biases. As this study aims to identify the genuine needs of residents within the context of human–land urbanization patterns, it is also essential to clarify during the training that the interviewees should be registered residents who have been residing in the town for a long period. Temporary residents or those who commute to the town for work do not fall within the scope of this investigation. The investigation adopts a semi-structured interview approach to avoid potential biases in questionnaire comprehension due to factors such as educational level. With the coordination and assistance of government officials, the investigators will visit residents' homes, explain the questionnaire to them, and collect data on an individual basis. This approach helps to ensure the validity of the questionnaire data to a certain extent."
In addition, to ensure adequate representation of the overall population, from the perspective of time, this study "selected both regular days and special days (weekdays: July 7-8, 2022; weekends: July 9-10, 2022)". In terms of the spatial distribution, this study "assigned investigators to conduct surveys in each residential area and balanced the data collection quantity between rural- and urban-biased residential areas." Furthermore, during the later stage of data processing, "8 questionnaires with missing or incomplete responses were excluded" to ensure data validity.
Point 2: The author(s) are recommended to change the subtitle of the article to the style of evidence from Zhejiang.
Response 2: Adding a subtitle to the manuscript to clarify the research area does contribute to its overall rigor. The title has been revised as follows:"Needs Hierarchy for Public Service Facilities and Guidance-Control Programming in Small Chinese Towns Caused by Complex Urbanization of Residents: The Style of Evidence from Zhejiang".
Point 3: The author(s) are recommended to describe their limitations and further directions separately.
Response 3: The limitations in the Conclusions chapter of this manuscript do need to be addressed more solidly. We have identified three limitations and provided further directions for each of them:"This study also has certain limitations. Firstly, as a pioneering study on the suitability of public service facilities programming in small towns, this study divided residential areas into two categories based on their degree of urbanization; namely, rural- and urban-biased areas. However, it did not address specific types of residential areas (limitation 1). Future studies may further explore the relationship between human–land urbanization characteristics of various types of residential areas and form a more comprehensive and refined facility programming system (further direction 1). Secondly, the selected sample in this study exhibits a matched urbanization pattern between human and land. Based on the results, it derives guidance-control schemes for 'required + optional' public service facility provisions under two mismatched scenarios: 'land > human' and 'land < human' (limitation 2). Future study should consider selecting samples that represent other human-land urbanization pattern mismatches to further validate and refine the guidance-control system (further direction 2). Thirdly, this study focused on the functional aspect of facility programming, addressing the question of 'what to construct.' However, a practical application process also involves the implementation of spatial aspects, addressing the question of 'where to construct,' (limitation 3) which requires cross-functional integration of the different types of residential areas and the determination of accessibility, as well as the corresponding life cycle of facilities (further direction 3). Therefore, the programming path for the suitability of public service facilities in small towns requires further exploration."
Point 4: Comments on the Quality of English Language:Minor editing of English language required
Response 4: We have submitted the manuscript to a professional editing service agency for language polishing, making corrections to inappropriate word choices and grammar errors.
The above has been revised the responses of the manuscript. Your careful review has provided more help to our study clearer and more comprehensive. Thanks again for the advice of expert!